# Broken Terrains v. 1.0: A supervised detection of fault-related lineaments on triangulated models of subsurface interfaces with preferred orientation

Michał P. Michalak[1], Christian Gerhards[2], Peter Menzel[2]

[1]Faculty of Geology, Geophysics and Environmental Protection, AGH University of Science and Technology, Mickiewicza 30, 30-059 Cracow, Poland, ORCID: https://orcid.org/0000-0002-1376-235X

[2]Institute of Geophysics and Geoinformatics, TU Bergakademie Freiberg, Gustav-Zeuner-Straße 12, 09599 Freiberg, Germany

*Correspondence to*: Michał P. Michalak (michalm@agh.edu.pl)

**Abstract.**

The study presents a novel approach for fault detection on geological terrains using supervised learning algorithm and careful input variables (features) selection. Synthetic faulted terrains are generated using Delaunay triangulation via the Computational Geometry Algorithms Library (CGAL) allowing for adjustments of parameters. We introduce 24 features, including local geometric features and neighbourhood analysis, for classification. Support Vector Machine (SVM) is employed as the classification algorithm, achieving high precision and recall rates for fault-related observations. Application to real borehole data demonstrates the effectiveness of the method in detecting fault orientations, the challenges remain with respect to distinguishing faults with opposite dip directions. The study highlights the need to address 3D fault zone complexities and their identification. Despite limitations, the proposed supervised approach offers significant advancement over clustering-based methods, showing promise in detecting faults of various orientations. Future research directions include exploring more complex geological scenarios and refining fault detection methodologies.

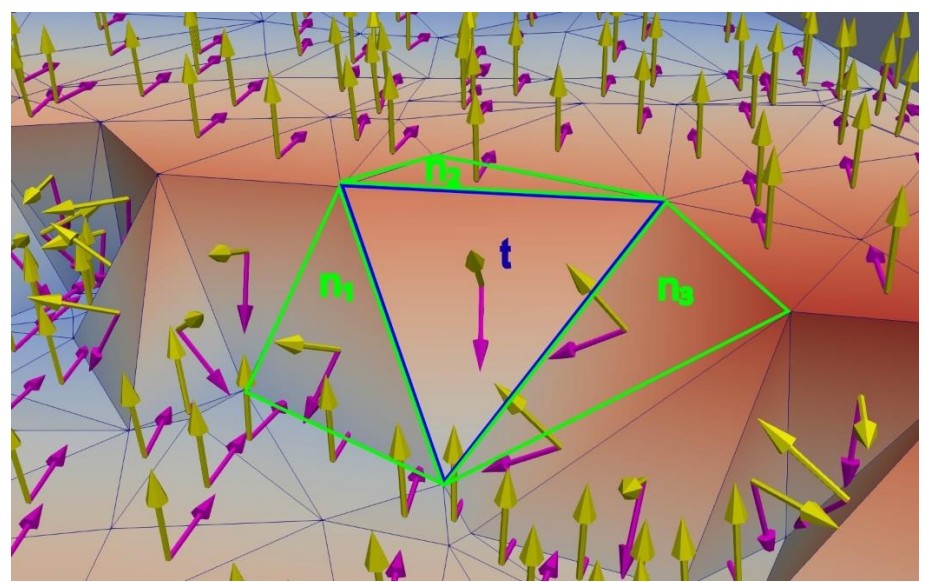

**Short Summary**

Using geometric features of synthetic triangulated models of geological terrains with preferred orientation, we applied machine learning to detect faults. Testing on real borehole data validated its effectiveness across various fault orientations. The supervised approach represents a significant improvement over older methods that relied on simpler clustering techniques which were capable of identifying less orientations of faults.

## 1 Introduction

Geological engineers and structural geologists aim to identify lineaments or faults on geological terrains. However, current methods are typically tailored for seismic data rather than terrains (An et al., 2021; Kaur et al., 2023). Additionally, supervised methods for fault detection can encounter challenges related to subjectivity, ambiguity, or time-consuming processes such as manual labeling of training data (Mattéo et al., 2021; Vega-Ramirez et al., 2021).

In this study, we focus on detecting faults on triangulated models of subsurface geological interfaces with preferred orientation (Fig. 1). The interfaces can be thought of as boundaries between conformal (sub-parallel) geological units often investigated in geological modelling (de la Varga et al., 2018). Data used in our study come from an irregular network of boreholes that document the transition between geological units. Consequently, the fault-related deviations from the preferred orientation are investigated using a supervised framework. In this approach, faces of the triangulation are observations described by features such as their orientation and geometric relationships with neighbours.

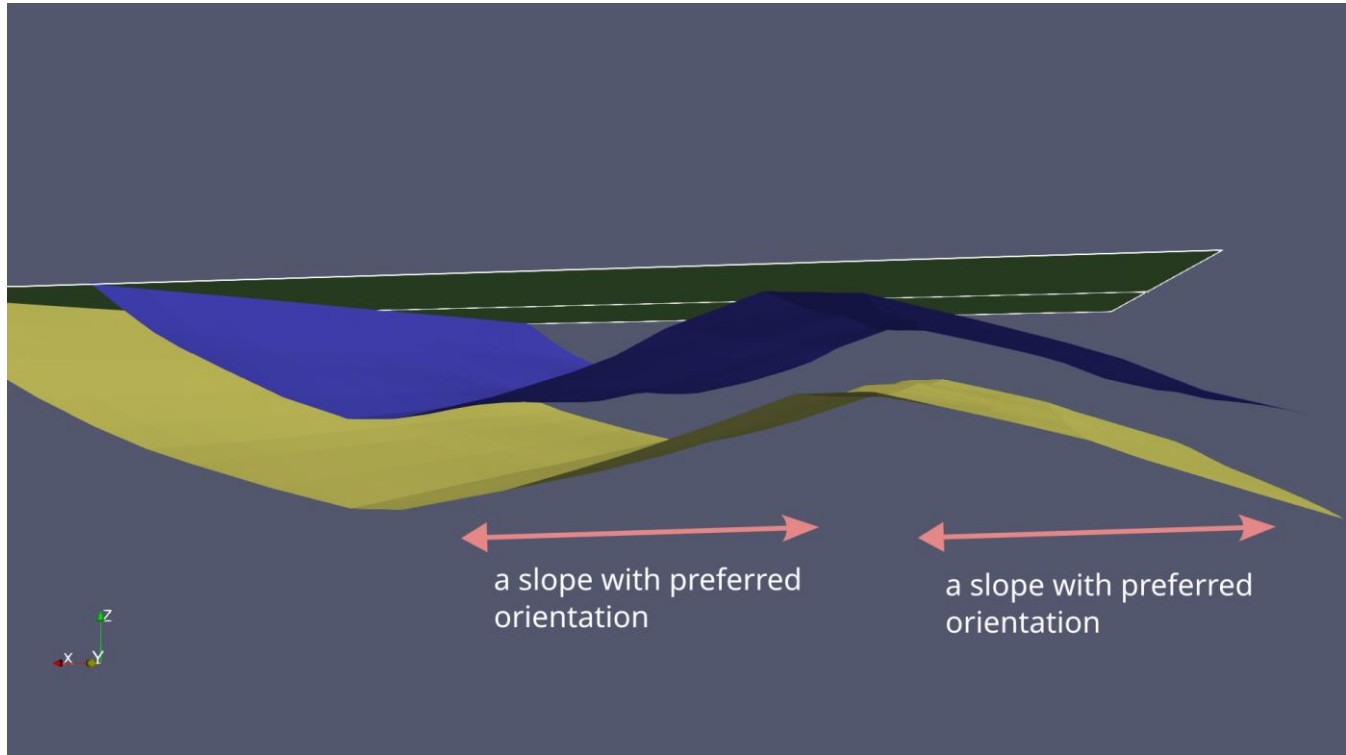

**Figure 1.** The supervised fault detection method developed in this study is designed for terrains with preferred orientation. An example of such a terrain can be a portion of the boundary between folded strata. The model with two folded surface and one erosive surface presented in this figure was generated using GemPy and GemGis software (Jüstel et al., 2022, 2023; de la Varga et al., 2018).

Our goal is to expedite the process of generating ground truth data for faulted triangulated terrains using the Computational Geometry Algorithms Library (CGAL.org, 2023). We will employ supervised machine learning algorithms for binary classification to predict possible fault presence within faulted terrains (Fig. 2). Our hypothesis posits that while traditional geometric attributes such as normal or dip vectors can still be useful for classification, integrating features reflecting angular relationships between triangles and their neighbours is crucial for accurate classification, especially for fault detection on homoclines. We assert that analyzing distances for neighbours (Fig. 3) is advantageous due to its insensitivity to terrain rotation, unlike traditional geometric attributes such as dip direction (Hu et al., 2021) or the orientation of normal vectors (Michalak et al., 2022). As such, neighbourhood analysis can be linked, e.g., to curvature in seismic data (de Oliveira Neto et al., 2023) in terms of its insensitivity to terrain rotation.

The main challenges relate to the effectiveness of machine learning algorithms, features selection, and the applicability of the method to diverse geological structures, potentially impacting classification accuracy and generalizability. To mitigate these challenges, we will conduct optimization of the algorithm's performance and features selection (see Methods). Validation across various geological terrains will ensure the method's robustness and applicability for fault detection on homoclines (Fig.

4). This structured approach aims to enhance classification accuracy and the method's utility in practical geological applications. The overall workflow of the study is presented in Fig. 5.

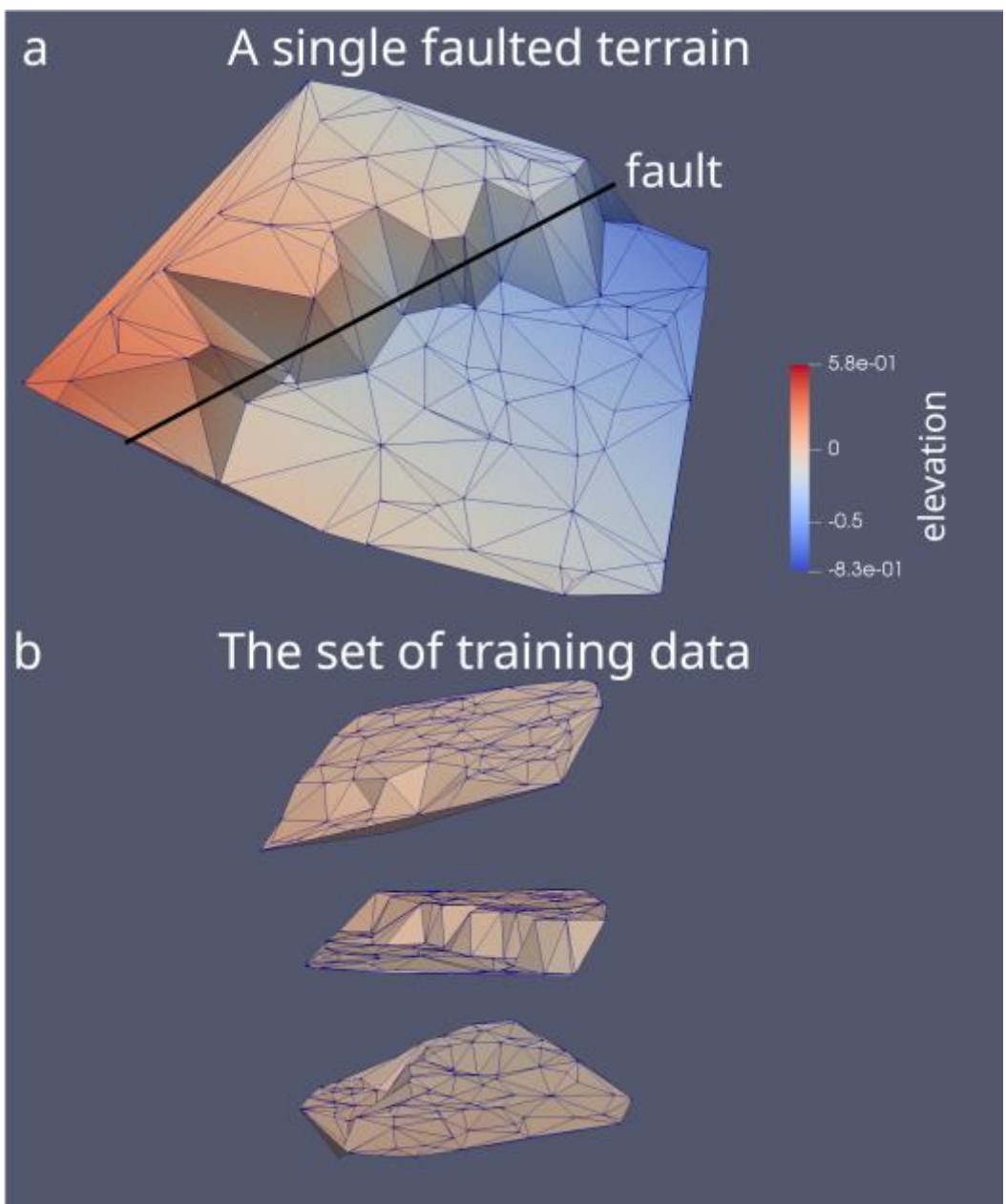

**Figure 2.** A triangulated model of a faulted geological terrain: (a) we can see an inclined terrain and triangles that intersect a
fault line. (b) A set of terrains with different parameters (dip angle and dip direction) can be used as training data in the classification task. In this panel, we showed only three terrains, but in practice an arbitrary number of terrains can be generated.

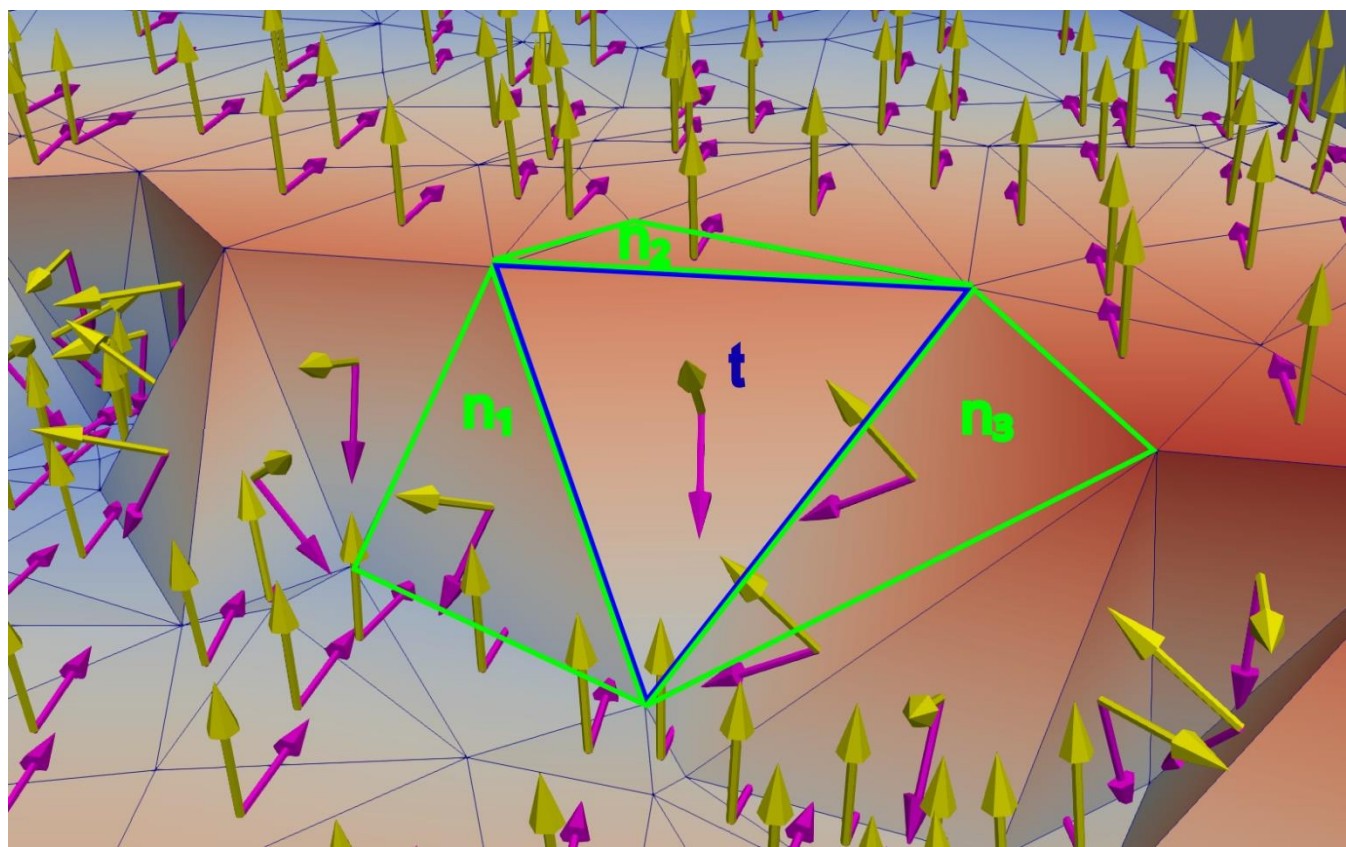

**Figure 3.** Because the fault introduced changes in the angular relationships between the orientation of fault-related triangles (t) and their neighbours (n1, n2, n3), the analysis of these relationships is essential for a successful classification. For example, we can measure the angular distance between normal vectors for three pairs corresponding to a specific triangle and its neighbours. The resulting value of angular distance can serve as a feature in the classification task.

## 2 Background

### 2.1 Related methodological developments

In geological mapping, machine-learning methods have been applied in the supervised lithology classification (Cracknell and Reading, 2014; Kuhn et al., 2018; Xiong and Zuo, 2021; Wang et al., 2020). In geological engineering, unsupervised methods were used to delineate subsets of observations representing discontinuities (Hammah and Curran, 1999; Zhan et al., 2017). In subsurface geological modelling, neural networks were used to delineate paleovalleys using topographic data as input data (Jiang et al., 2021) and convolutional neural network were used to create geological models with structural features controlled by a set of random parameters (Bi et al., 2022). As a specific unsupervised learning method, clustering algorithms find application in triangulated geological terrains (Michalak et al., 2022). For example, the k-means algorithm generates partitions comprising geometrically similar observations based on cosine similarity (Choi et al., 2014). However, they place the burden on the user to determine whether a specific observation represents a fault. This can pose challenges, as some anomalous orientations may be associated with other structures or measurement errors. Moreover, applying unsupervised learning to 3D orientations reveals sensitivity to the choice of vectorial representation (Michalak et al., 2022), resulting in varying clustering results for dip and normal vectors. From a geological viewpoint, a serious limitation of unsupervised learning is that the orientations of the identified lineaments depend on the partition resulting from the clustering. For example, previous results suggest that for uniformly oriented sub-horizontal terrains (homoclines) the clustering algorithms may find it difficult to distinguish between observations related to the regional trend and observations related to faults striking perpendicular to the regional trend (Fig. 4b, 4c) (Michalak et al., 2022). In the problem of fault or lineament detection, the majority of available supervised methods are primarily tailored for seismic data (An et al., 2021; Kaur et al., 2023). For topographic data, supervised methods were utilized for fault-scarp prediction (Vega-Ramirez et al., 2021) using Fisher Linear Discriminant Analysis. However, this analysis relied on high-resolution bathymetric data based on a small training dataset (163 samples). Another example involves the use of topographic attributes such as DEM, slope, aspect, faults and environmental features such as vegetation and climate for monitoring of ground deformation (Hu et al., 2021).

## 2.2 Geological Setting

### 2.2.1 Regional and geometric background

As a relevant case study, we selected Kraków-Silesian Homocline (KSH) – a geological unit considered to be a slope of the Szczecin-Łódź-Miechów Synclinorium. The formation of KSH is mainly attributed to the inversion of the Permian-Mesosoic Polish Basin (Dadlez et al., 1995; Słonka and Krzywiec, 2019). From a geometric perspective, KSH dips at low angles to NE (Matyszkiewicz et al., 2015; Marynowski et al., 2007; Michalak et al., 2019; Znosko, 1960). It is generally assumed that the faults form a unimodal set of sub-parallel faults trending NE-SW (Fig. 4a) (Hermański, 1993; Bardziński et al., 1985). However, later clustering experiments with two or three clusters (Michalak et al., 2022) added knowledge about geometric anomalies also aligned with the N-S direction (Fig. 4b, 4c).

### 2.2.2 Discussion of previous results

Little is known about faults trending perpendicular to the preferred dip direction. While the results (Fig. 4b, 4c) suggest that they may not exist, we note that this negative effect can be due to limitations of unsupervised learning methods: the spatial distribution of labels depends on the partition induced by clustering algorithms. This dependence may result in visual disintegration of rare structures represented by observations being in different clusters. For example, the boundary between blue and purple labels may be related to faults dipping to SW (Fig. 4). Likewise, it is unlikely that all observations dipping to NE are genetically related to the homocline; instead, observations dipping to NE but having dip angle greater than that of the homocline, may be related to faults dipping to NE.

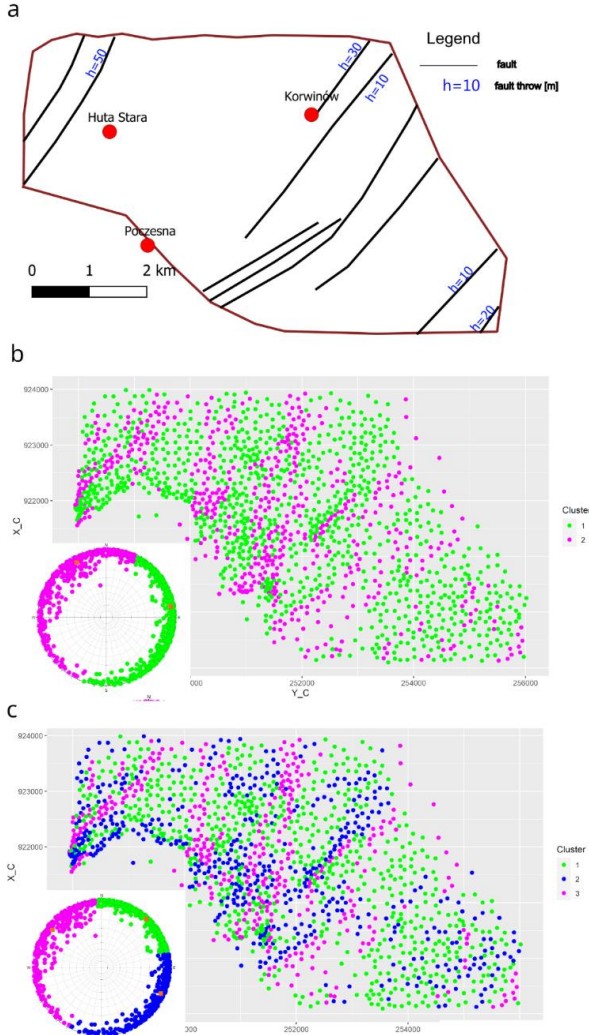

**Figure 4.** Progressing knowledge about tectonics of the Kraków-Silesian Homocline. (a) due to abandoned mining activity in the area it was possible to confirm some of the faults and their properties such as fault throw in underground mines (Hermański, 1993). Later experiments based on cluster analysis (k-means algorithm, Michalak et al., 2022) of normal and dip vectors provided evidence about the orientation of geometric anomalies. (b) – clustering of dip vectors for two clusters. The spatial distribution of labels suggests presence of geometric anomalies trending from S-N to SW-NE (c) – clustering of dip vectors for three clusters. The spatial distribution of labels in the NW part of the study area suggests presence of more than one fault trending SW-NE with opposite dip direction. However, the partition induced by the clustering makes it impossible to identify faults dipping to NE steeper than the homocline. Data used to generate labels in panels (b) and (c) are borehole data used in this study, as well. The figure is a modified figure from using unsupervised classification of triangulated models of borehole data used in this study (Michalak et al., 2022).

**3 Methods**

In this section, we explain the workflow starting from generating synthetic data, through training the model on synthetic data to the final evaluation on real data. The stages are summarised in Fig. 5.

**3.1 Generating terrains**

In our study, the synthetic training data consists of one thousand triangulated terrains using the Delaunay triangulation (De Berg et al., 2008). A user has the flexibility to adjust parameters of the resulting data set in the following fields: the number of files to generate, the lower and upper bound of terrain sizes, the left and right range of the dip direction, the lower and upper bound of the dip angle, the lower and upper bound of the number of points in the triangulation, the lower and upper bound of the surface noise, the lower and upper bound of the fault throw. The user-defined parameters and their admissible ranges are specified in Table 1. However, if a constant value of a parameter should be investigated, it is possible to set the same value for the lower and upper bound. In the next step, tools from the C++ random number library generate random numbers from the uniform distribution using the bounds entered by a user. The information about parameters is saved to a text file to allow further inspection.

The faulted triangulated terrains are created in the following sequence which is also summarized in Fig. 6 (numbers 1-6 below correspond to letters e-f in Fig. 6).

1. A container with 2D points is generated within a square of a given size.
2. A new container of 3D points is created with the Z coordinate corresponding to the random value of dip and dip direction (ranges specified by a user).
3. Noise is introduced to the surface defined as a random fraction (ranges specified by a user) of the elevation difference within the terrain.
4. A fault is introduced with the throw defined as a random fraction (ranges specified by a user) of the maximum elevation difference within the generated terrains. The orientation of the fault is determined by two points randomly selected from the boundary of the square.
5. Triangulation of the terrain is performed and the attributes including relationships with neighbours are calculated.
6. Classification task involves labeling each observation based on whether it is a fault-related observation (label=1) or not (label=-1). Therefore, we use the intersection predicate (CGAL.org, 2023) to test whether a specific triangle intersect the line representing a fault.

Following this approach, we are capable of generating great amount of synthetic and labeled ground truth data. Parameters used in this study are given in the Table 2. To ensure that the training is performed on good quality data, we removed triangles with high degree (0.90 and greater) of collinearity defined as a ratio between the longest triangle's edge and the sum of remaining lengths (Michalak, 2018). This coefficient lies in the interval [0.5, 1] with lower and higher values pointing to equilateral and collinear configurations, respectively (Michalak et al., 2021; Michalak, 2018).

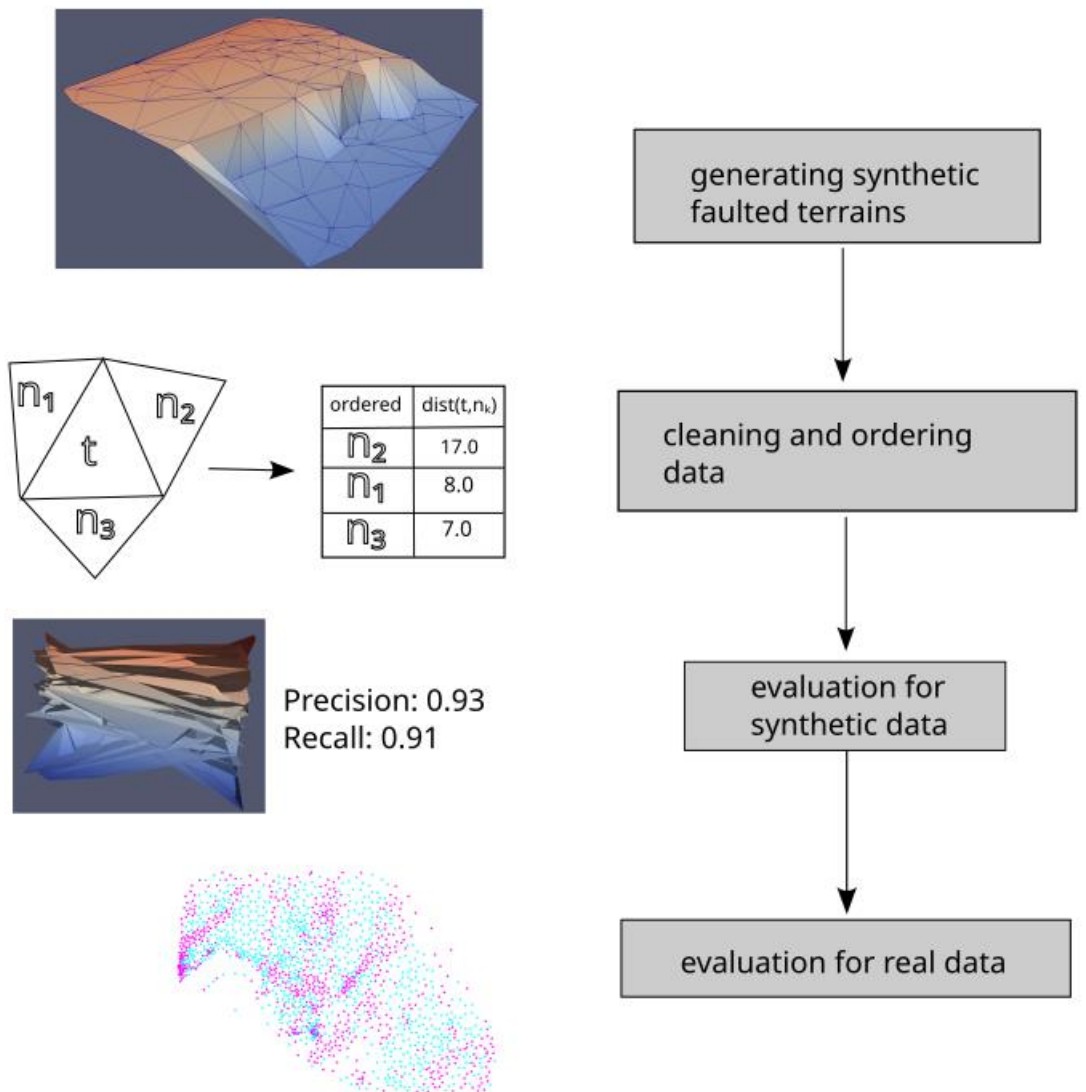

| ordered | dist(t,n_k) |
|---|---|
| $n_2$ | 17.0 |
| $n_1$ | 8.0 |
| $n_3$ | 7.0 |

Precision: 0.93
Recall: 0.91

**Figure 5.** Workflow applied in this study. We create one thousand random faulted geological terrains controlled by random parameters. Then, for each triangle, we sort the distances between neighbours to reduce randomness (see section 3.2 for a more detailed explanation). In the next step, we test the machine learning algorithm for synthetic data. At the end of the procedure, we evaluate the proposed approach for 165  real data to test generalizability.

**Table 1**. Input parameters. The parameters given in the table are specified by the user by providing ranges. To generate a single model, a number from the uniform distribution is drawn using the provided ranges. Please note that dip angle and dip direction are not used directly in the later stages as features for classification but first are converted into normal and dip vectors.

| Name of the parameter | Range of possible values | Name of the variable in the code |
|---|---|---|
| Number of files | >=1 | number_of_files |
| Terrain sizes | >0 | left_terrain_size, right_terrain_size |
| Dip angle | 0-90 | min_terrain_dip, max_terrain_dip |
| Dip direction | 0-360 | left_range_azimuth, right_range_azimuth |
| Number of points in the triangulation | >=3 | left_number_triangulation, right_number_triangulation |
| Noise of the surface | 0.00-1.00 | left_surface_noise, right_surface_noise |
| Fault throw | 0.00-1.00 | left_fault_throw, right_fault_throw |

**Table 2.** Values or ranges for parameters used in this study.

| Name of the paramter | Value |
|---|---|
| Number of files | 1000 |
| Terrain sizes | 1 |
| Dip angle | 0.5-2.0 |
| Dip direction | 20-70 |
| Number of points in the triangulation | 100 |
| Noise of the surface | 0.02-0.04 |
| Fault throw | 0.05-0.25 |

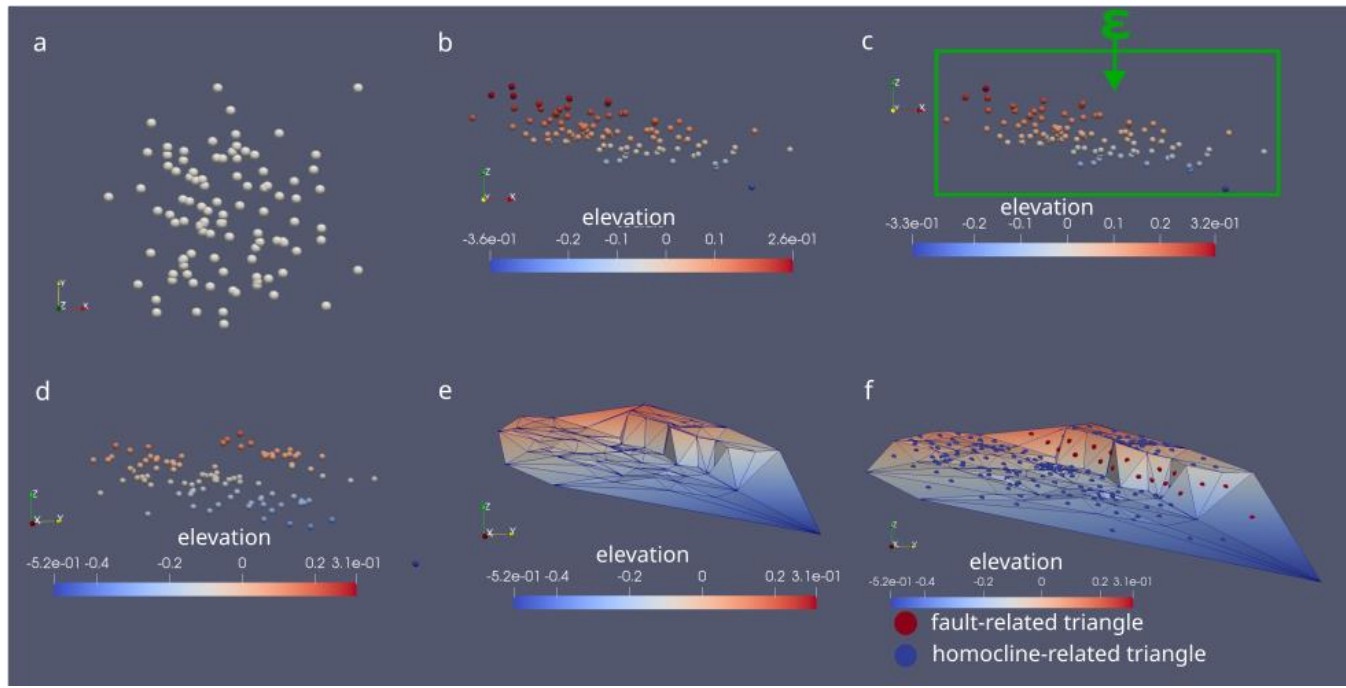

**Figure 6.** Depiction of sequence of processes applied to generate training data: (a) creating points in 2D space. (b) assigning elevation to the data depending on the randomly generated dip angle and dip direction, (c) adding noise to the data, (d) introducing faults and resulting elevation changes, (e) applying triangulation to the data, (f) labelling the data according to the intersection test with the fault line.

## 3.2 Selecting meaningful and consistent features

There can be numerous geometric features used for the purpose of classification such as dip angle or dip direction (Hu et al., 2021; Wang et al., 2021). However, including dip direction for classification as a value within the [0, 360] range may not always be successful. This is because northern directions indicate great numerical difference (e.g. 358-2=356) but very small geometric difference (4 degrees). Sometimes the limitations of using dip direction are acknowledged and the feature is removed from the analysis (Yang et al., 2023). Therefore, in this study we didn't use dip direction as a feature for classification.

In this study, to predict the correct label (label=1 for fault-related observations and label=-1 for non-fault-related observations), we used 24 features. The set consists of six local geometric features and eighteen features corresponding to the neighbourhood analysis. The first group consists of coordinates of normal and dip vectors. The second group includes features corresponding to the neighbourhood component of the analysis. The features are as follows: angular distance, Euclidean distance and cosine

distance applied to both normal and dip vector representations. The formulas for angular, Euclidean and cosine distances are given in the below equations, respectively:

$$d_a(x, x') = \text{acos}\left(\frac{|x \cdot x'|}{||x|| * ||x'||}\right) \tag{Eq. 1}$$

where "$\cdot$" is the dot product, and $||x||$ is the length of the vector $x$. In our case, the vectors have unit length. The use of absolute value in the numerator reflects the use of acute angles between vectors.

$$d_e(x, x') = ||x - x'|| \tag{Eq. 2}$$

$$d_c(x, x') = 1 - x \cdot x' \tag{Eq. 3}$$

However, in relation to the proposed neighbourhood analysis, an obstacle arises in processing this data due to the lack of a clear distinction between first, second, and third neighbouring triangles (Fig. 3, middle part of Fig. 5). This lack of order introduces randomness or arbitrariness into the analysis and compromises the consistency of data processing, which is crucial for the accuracy and reliability of the results.

To address this, we sort the distances to neighbouring triangles in decreasing order. Sorting these values eliminates randomness
from the analysis and ensures consistency in data processing, thereby enhancing the correctness and credibility of the results. Subsequently, several classification algorithms available in the scikit-learn library (Pedregosa et al., 2011) can be tested in terms of precision and recall. However, in this study we work with a single algorithm to keep focus on the new classification method. We selected the Support Vector Machine which is considered a suitable tool for binary classification problems in high-dimensional spaces (Bishop, 2006) and which performed well in terms of precision and recall in our preliminary research.

**3.3 Support Vector Machine**

For the purpose of binary classification, we used the support vector machine algorithm, a two class classifier (Bishop, 2006; Vapnik, 2000), available in the scikit-learn library (Pedregosa et al., 2011). The support vector machine algorithm can be considered an optimization algorithm because the decision is based on a hyperplane with the maximum margin. The margin is defined to be the minimal distance between a point in the training set and the hyperplane. The motivation behind the concept
of margin is that if a margin is large, then it will be capable of separating the training set even after small perturbation of the instances (Shalev-Shwartz and Ben-David, 2013). Formally, the optimization objective looks as follows (Bishop, 2006):

$$arg \max_{w,b} \left\{ \min_{n} \left[ t_n \left( \frac{w^T f(x_n) + b}{||w||} \right) \right] \right\}, \tag{Eq. 4}$$

where $t_n \in \{-1, 1\}$ are target values, $f(x)$ denotes a fixed feature-space transformation. This transformation is expected to facilitate separation of instances which were not linearly separable in the original space. Common choices of transformations
(kernel functions) include: linear, polynomial and radial basis functions. Next, $w$ is the vector of weights which determines the orientation of the decision surface, and $b$ is the bias parameter (not to be confused with bias in the statistical sense). The expression $\left( \frac{w^T f(x_n) + b}{||w||} \right)$ denotes the perpendicular distance of a point $x_n$ to the decision surface $y(x) = w^T f(x) + b = 0$. This decision surface separates points with different labels: $-1$ and $1$. The multiplication $t_n y(x_n)$ visible in the optimization

task filters solutions for which all data points are correctly classified, i.e. $t_n y(x_n) > 0$ . We note that in some formulations of the optimization problem, the vector of weights has unit length (Shalev-Shwartz and Ben-David, 2013). Because all $N$ points lie beyond the margin area, they are at some distance from the hyperplane corresponding to the size of the margin. While the distances of $N$ points relative to the decision boundary can be different, they are all greater than a fixed number corresponding to the size of the margin which can be expressed by a set of $N$ inequalities. Therefore, the optimization objective (Eq. 4) together with the set of $N$ inequalities form a constrained optimization problem which can be solved be using the Lagrange multipliers (Bishop, 2006 - Appendix E).

In the presence of outliers, a soft margin classifier can be applied which allows some samples to be classified incorrectly (Shalev-Shwartz and Ben-David, 2013). For the radial basis function kernel, C and gamma parameters are considered. The parameter C, common to all SVM kernels, is a penalty parameter: a low C tends to make the decision surface simple (thus, avoiding overfitting but possibly affecting the correct classification of the training data), while setting a high C will result in classifying training examples more correctly (possibly leading to poorer generalizability). Gamma defines the radius of the similarity of a single training sample. The lower gamma is, the greater the similarity radius of a sample (Pedregosa et al., 2011).

We use the following metrics as evaluation metrics:

$$precision = \frac{true\ positive}{true\ positive + false\ positive} \text{ and } recall = \frac{true\ positive}{true\ positive + false\ negatives} .$$

The definition implies that precision is maximized if there are no false positives and the recall is maximized when there are no false negatives. Based on these definitions the harmonic mean of both can be defined as follows

$$F_1 = 2 * \frac{1}{\frac{1}{precision} + \frac{1}{recall}} = 2 * \frac{precision * recall}{precision + recall} .$$

### 3.4 Spatial clustering

In our study, we visualize the classification results for real data using the concept of spatial clustering (Fisher, 1993; Fisher et al., 1985). The definition of spatial clustering is studying first the directional information of the data without taking into account spatial information. This study aims to group geometrically similar observations and at the end the resulting clusters of directions are put back into their spatial context (Fisher, 1993). Indeed, in our case the Support Vector Machine algorithm performs the classification ignoring spatial information. It uses only geometric information such as the orientation of a triangle and the relationships between a triangle and its neighbours. The labels of clusters grouping similar triangles are recorded initially as integers corresponding to fault-related triangles (label=1) or triangles belonging to the homocline (label=-1). Then, the integers are converted to colors and presented on a map receiving again spatial information. As Fisher notes (Fisher, 1993), the spatial clustering is somewhat paradoxical: "to perform the desired directional-spatial clustering, it may be necessary to decouple the directional from the spatial information initially".

**4 Results**

**4.1 Synthetic data**

In our study, we used 1000 triangulated terrains with 100 points in every terrain (see also Table 2). This configuration resulted in the initial number of 185980 triangles. We removed collinear configurations (collinearity>0.90) and triangles which did not have three finite neighbours. As a result, 145297 triangles remained. And only a small fraction of triangles are fault-related triangles (12411 vs 132886). Therefore, to reduce class-imbalance, we randomly select 12411 observations from the class with

255 non-fault observations. Taking all considerations into account, we have 24822 samples with 12411 observations for each class (-1 and 1). Then, the set was divided into training (18616) and test (6206) set.

For arbitrarily selected hyperparameters (C=0.05, gamma=0.042), with radial basis function as kernel function) in the scikit-learn framework we achieved the following confusion matrix for the test data (Table 3):

**Table 3.** Confusion matrix for the classification task before hyperparameter tuning. The sum of the entries is equal to the number of samples in the test data.

| 2993 (true negatives) | 243 (false positives) |
|---|---|
| 123 (false negatives) | 2847 (true positives) |

The values of precision and recall for the fault-related observations are 0.92 and 0.96, respectively (Table 4). However, the arbitrarily selected hyperparameters are not guaranteed to give the best performance of the algorithm. To further increase the

265 values of the classification metrics, we tested many combinations of the hyperparameters as a part of the grid search optimization (Pedregosa et al., 2011). The optimal combination of hyperparameters turned out to be as follows: C=10, gamma=0.01 with radial basis function as the kernel function. The classification results change slightly after the grid optimization stage (Tables 5 and 6). We note that after every execution of the code the values of the hyperparameters can change due to random steps of the procedure.

**Table 4.** Results for the classification of test data (unseen terrains) for arbitrarily selected hyperparameters (before hyperparameter tuning)

| Class | Precision | Recall | F1-score |
|---|---|---|---|
| Non-fault | 0.96 | 0.92 | 0.94 |
| Fault | 0.92 | 0.96 | 0.94 |

**Table 5.** Confusion matrix for the classification task after hyperparameter tuning. The sum of the entries is equal to the number of samples in the test data.

| 2982 (true negatives) | 163 (false positives) |
|---|---|
| 134 (false negatives) | 2927 (true positives) |

**Table 6.** Results for the classification of test data (unseen terrains) after fine-tuning of the hyperparameters during grid search

optimization

| Class | Precision | Recall | F1-score |
|---|---|---|---|
| Non-fault | 0.96 | 0.95 | 0.95 |
| Fault | 0.95 | 0.96 | 0.95 |

## 4.2 Real data

To investigate generalizability of the method for real data, we used borehole data (Michalak, 2024a) corresponding to a horizon

separating Middle-Jurassic rock units: Aalenian-Early Bajocian Kościeliska sandstones from Late Bajocian-Late Bathonian ore-bearing clay deposits (Matyja and Wierzbowski, 2000; Kopik, 1998) (see also the section 2.2 Geological setting).

The results of supervised classification using SVM are similar to those obtained using unsupervised (Michalak et al., 2022) classification (compare Figs 4 and 7) in that the majority of faults has the SW-NE, SSW-NNE or S-N orientation. However, there are significant differences which relate to visibility of new potential faults trending perpendicular to the preferred dip

direction. For example, Fig. 7a in the central part (near coordinates 921500, 251000) shows two potential faults trending NW-SE at the termination of S-N and SSW-NNE trending potential faults. Another difference is that the unsupervised classification presented the major fault in the NW part of the study area as possibly composed of smaller faults with opposite dip direction (Fig. 4c, near coordinates 922000, 248500). In contrast, the binary classification cannot distinguish between faults with opposite dip directions. Therefore, the zone of fault-related labels near the discussed fault zone appears relatively wide.

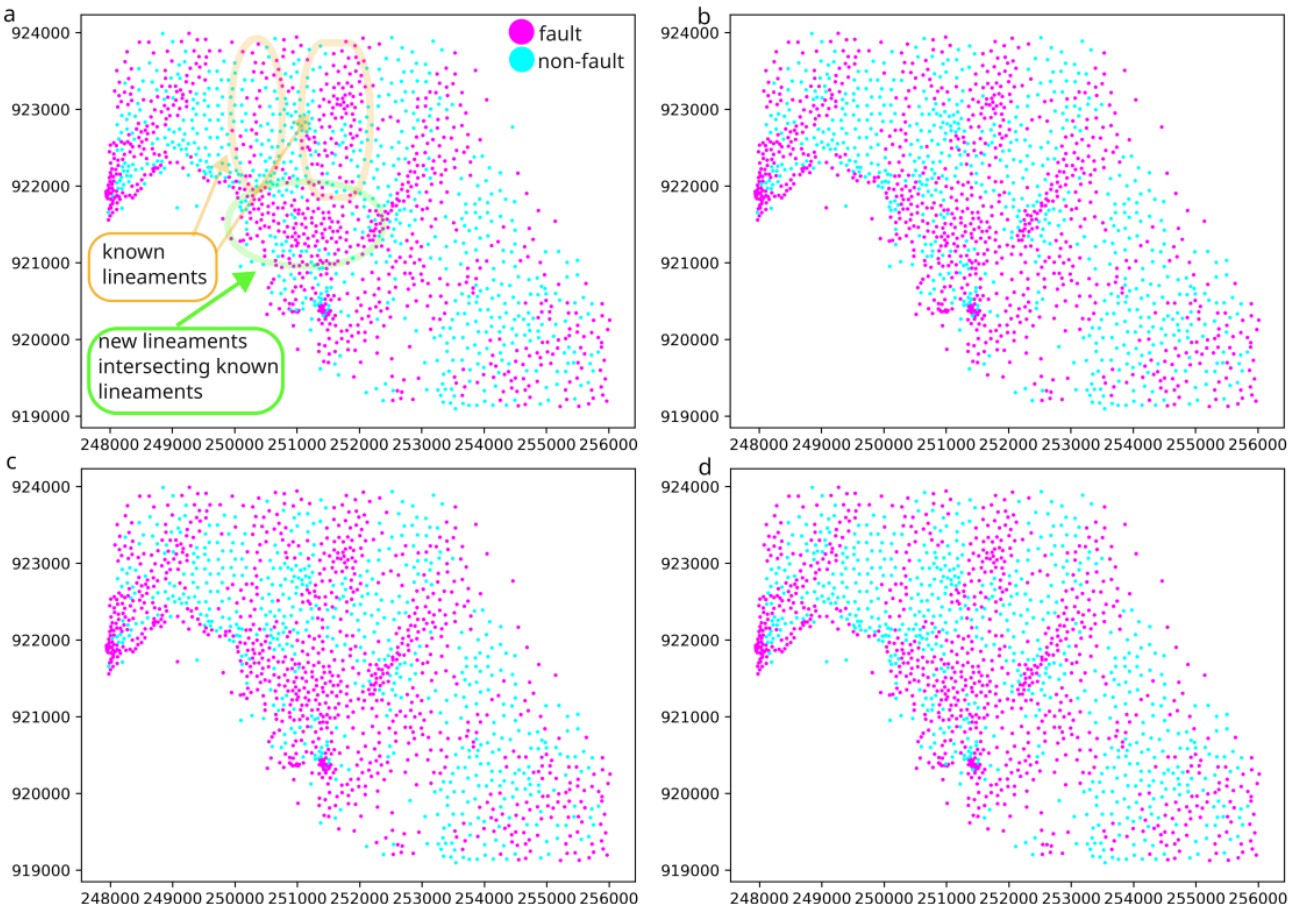

**Figure 7.** Classification results for the Kraków-Silesian Homocline: (a) the optimal combination of the hyperparameters as suggested by the grid search optimization (C=10, gamma=0.01, with radial basis function as the kernel function) (b) a custom combination of the hyperparameters (C=1, with linear kernel) (c) a custom combination of the hyperparameters (C=10, with linear kernel) (d) a custom combination of the hyperparameters (C=10, gamma=0.01 with radial basis function as the kernel function). On the panel (a), we marked lineaments known from the unsupervised learning revealed in previous studies (Michalak et al., 2022). The new lineaments revealed in this study using the supervised classification and the area where both groups intersect are marked, as well.

## 5 Discussion

### 5.1 Advantages of using faces of triangulation

In our study, we used local geometric attributes of triangles and neighbourhood analysis to predict faults on triangulated models of subsurface geological terrains with preferred orientation. In subsurface geological modelling, the neighbourhood analysis was already applied for individual boreholes of triangulated surfaces to analyze connectivity of strata (Guo et al., 2024). From a viewpoint of graph theory, in our case the neighbourhood analysis is performed on finite faces of the triangulation rather than on its finite vertices (boreholes). Because, for every triangulation, with $k$ being the number of points on the edge of the convex hull, the relationship between vertices ($n$) and triangles ($m$) is $m = 2n - 2 - k$ (De Berg et al., 2008), our approach will usually (except very small data sets) result in a greater number of observations compared to a potential approach of considering boreholes as observations. Moreover, our approach ensures that every observation has three finite neighbours which testifies that observations are comparable. When neighbours of points are considered, this is not the case because the degree of a vertex usually is not a constant number.

### 5.2 Comparison with unsupervised approaches

The improvement of the supervised approach over the unsupervised method is that the clustering results depend on the partition generated by clustering algorithms. Therefore, the unsupervised version offers to examine spatial distribution of clusters. But it doesn't offer the examination of spatial distribution of structures related to different clusters. Moreover, the empirical results showed that clustering algorithms often struggle to separate regional trend from faults striking perpendicular to the regional trend on homoclines (Fig. 4b, 4c). Compared to the unsupervised method (Fig. 4c, the NW part of the study area), the main drawback of the supervised approach is that the algorithm cannot distinguish between different dip directions of a fault. Therefore, a zone of fault-related labels may consist of many sub-parallel sequences of labels corresponding to more than one sub-parallel faults possibly with opposite dip direction (Fig. 7, the NW part of the study area).

### 5.3 Complexities of real data

The borehole data set documenting the interface between Kościeliska sandstones and ore-bearing clays has been traditionally used for inferring tectonics (Znosko, 1960). However, it should be admitted that there is a hiatus covering the earliest Late Bajocian confirmed by the lack of Strenoceras subfurcatum Ammonite Zone (Garbowska, 1978). It is unclear whether some of the identified lineaments (Fig. 7) can be attributed to erosion. For example, some researchers point out that erosion took place during the earliest Late Bajocian (Dayczak-Calikowska and Moryc, 1988). However, some underground observations did not confirm deviations from the general parallelism between older Kościeliska sandstones and younger ore-bearing clays.

Moreover, the orientation of the interface separating Kościeliska sandstones from ore-bearing clays was assumed to be uniformly inclined to North East in hydrogeological models during exploitation (Hermański, 1971).

### 5.4 Modelling assumptions

The main assumption for generating the terrain data (section 3.1) is that a fault is always represented by a plane. In our case,

we assume that any terrain point lies either on one or on the other side of the fault and it is not possible that any terrain point is located on the fault. When these terrain points are triangulated, each fault-related triangle connects points from both sides of the fault. Every fault-related triangle has at least one neighbour that is not associated with the fault. The fault is represented by a stripe of the fault-related triangles (see triangles with green markers in Fig. 8A).

This main assumption is a simplification. In reality, a fault structure is never a purely planar object. It is mostly a 3D structure

that has an extension perpendicular to the fault line direction (Childs et al., 2009). Therefore, it is possible that at least some terrain points may be located inside the fault zone. When triangulating these points, there is the possibility to create fault-related triangles all of whose neighbours are fault-related triangles as well. (see Fig. 8B, triangles with red markers).

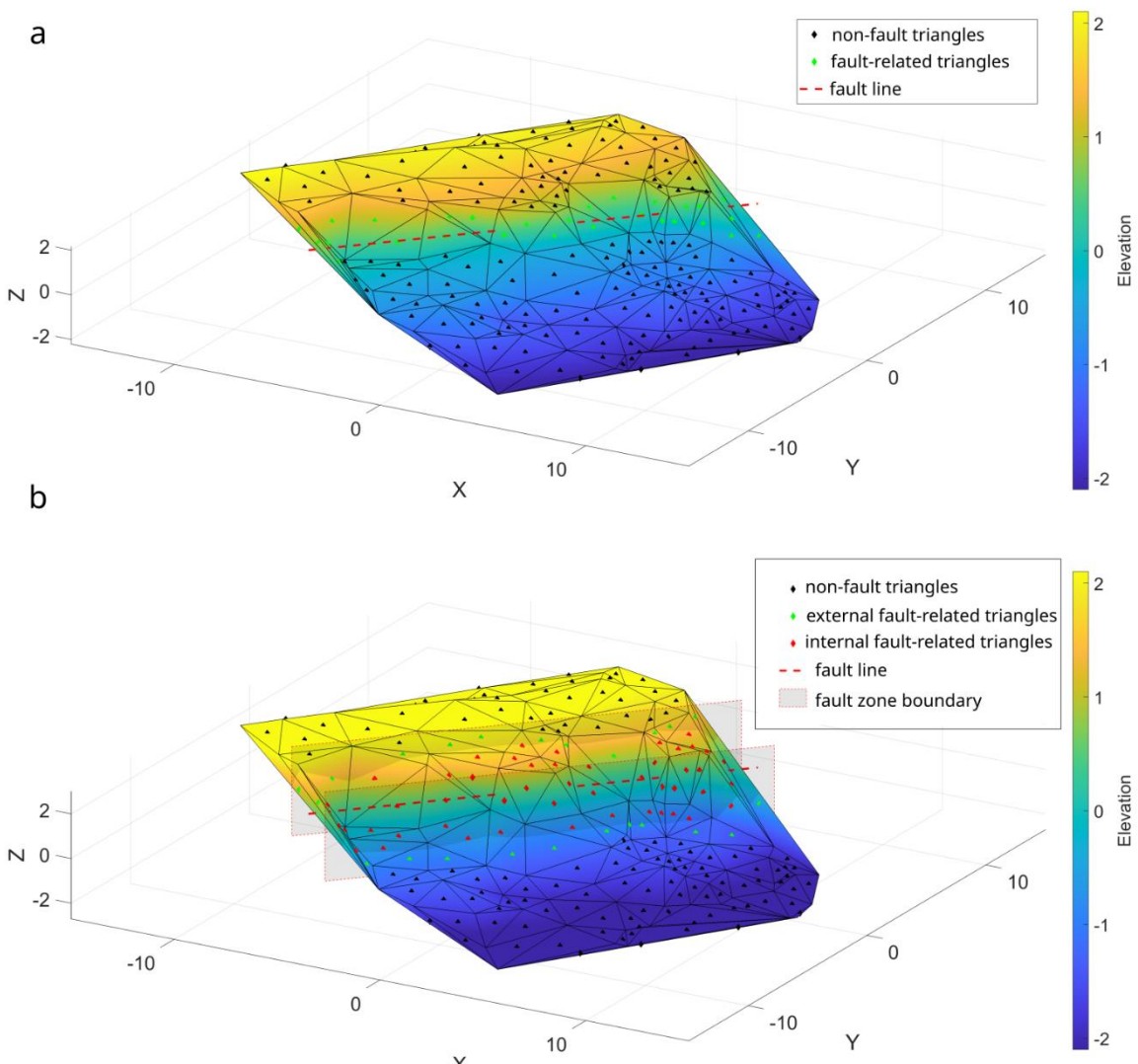

**Figure 8.** (a) Example for a terrain generated as explained in the section 3.1 with the fault being represented by a plane. (b) Example for a terrain generated as explained in the section 3.1 with extended fault zone. Triangles with green markers show at least one non fault-related triangle as a neighbour. Triangles with red markers have only fault-related triangles as neighbours. Grey planes represent the border of the extended fault zone.

These "internal fault-related" triangles show a combination of patterns that was never trained. Triangles that exhibit patterns that were actually trained are still present (see Fig. 8B, triangles with green markers) but are only located at the borders of the fault zone (grey rectangles in Fig. 8B). This leads to the assumption that the presented classification system may classify the extended fault zone not by one sequence of color-coded labels but by two quasi-parallel sequences of this type. Whether the "internal fault-related triangles" can be successfully classified is not clear. In terms of local geometric features specific to a single triangle such as coordinates of normal vectors, the "internal fault-related" triangles are more similar to the classical fault-related triangles. In contrast, the neighbourhood analysis alone would likely classify these triangles as non fault-related triangles (triangles with black marker in Figs. 8A and 8B).

The influence of 3D fault zones for the classification result needs to be further studied. Nevertheless, in the context of this study the data points are assumed to be sparsely scattered. If the fault shows an extension significantly lower than the mean data point spacing, the simplified assumption of purely planar faults is valid. The problem of identifying internal fault triangles could possibly be ameliorated by including not only the direct neighbours of a triangle in the training procedure but also triangles that are $2^{nd}$-, $3^{rd}$-degree neighbours or even higher degrees. Using this approach, it is more likely that the extended neighbourhood of an "internal fault-related" triangle also contains non-fault related triangles. However, the concept of the suggested approach would stay the same. For illustrational purposes we, therefore, stick to the simplest setup.

## 6 Conclusions

In this study, we developed a supervised fault detection method for triangulated models of geological terrains with preferred orientation. The novelty lies in generating large amount of synthetic terrains using the CGAL library and their triangulated models using Delaunay triangulation. The orientation of individual triangles combined with geometric relationships with neighbours are used as features for classification. The proposed supervised method has the potential to identify fault-related lineaments of any orientation which can be considered improvement over unsupervised classification approaches. The main challenge of the workflow is to eliminate arbitrariness in features selection in relation to neighbourhood analysis. Sorting distances among neighbours eliminates arbitrariness from the analysis but it is also the most computationally intensive part of the workflow. We believe that the classification approach can be used by geologists interested in geological complexity of subsurface environments with limited availability of data. Further studies can focus on considering more complex geological scenarios including the influence of 3D fault zones and physics-based models (compare with Conclusions in Reichstein et al., 2019).

## Code availability

**Name of code**: BrokenTerrains. **License**: GNU General Public License v3.0. **Developer**: Michał Michalak. **Contact address**: AGH University of Krakow, Poland. E-mail: michalm@agh.edu.pl . **Year first available**: 2024. **Hardware required**: The computer code was run on a laptop with Intel(R) Core ™ i7-7500U CPU 2.70 GHz, 16 GB RAM. **Software required**: CGAL library (v. 4.8), Microsoft Visual Studio 2022. **Program language**: C++, Python. **Program size**: 738  KB. **How to access the source code**: (Michalak, 2024b) https://doi.org/10.5281/zenodo.12375568 Setup guide: https://github.com/michalmichalak997/BrokenTerrains/blob/main/README.md

## Data availability

Datasets for this research (input and processed data) are available in these intext data citation references (Michalak, 2024a)

## Author contribution

MM devised the project, wrote the computer code and the manuscript, performed the computations and discussed the results. CG participated in the study conceptualization (sorting distances with neighbours), PM discussed the results (Modelling assumptions).

## Competing interests

The authors declare that they have no conflict of interest.

## Acknowledgements

Research project supported by program „Excellence initiative – research university" for the AGH University. AI tools (ChatGPT) were used for improving computer code files and the English language. I thank Tomasz Zając from Bluemetrica for initial discussions about formulation of the supervised fault-detection problem.

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
