# Peer review of "Broken Terrains v. 1.0: A supervised detection of fault-related lineaments on triangulated models of subsurface interfaces with preferred orientation"

_EGUsphere, 2024_

## Author Comment (AC2)

In this file, there are responses to both Reviewer#1 and Reviewer#2.

**Responses to Reviewer #1**

We thank the Reviewers #1 and Reviewer #2 for submitting their reviews in due time and for constructive comments. We agree that explicitly stating the assumptions can be helpful to better follow the manuscript. However, we were unable to answer some of the comments from Reviewer #1 easily:

- comment #15 - we don't understand the symbols "XXXX"
- comments #2 and #25 say that Figures 2 and 3 are illustrating steps in the study but only Figure 3 does illustrate steps.
- comment #14 - The question is mathematically incomprehensible because a vector does not have its own distance ("distances of normal vectors").

To better observe communicating our response, we divided our responses into three categories: Agree/Clarification/Disagree.

1.

| Suggestion, Question, or Comment from the Reviewer #1 | Author's Response | Change in the Manuscript |
|---|---|---|
| The manuscript titled "Broken Terrains v. 1.0: A supervised detection of fault-related lineaments on geological terrains" describes an approach to predict fault 3D geometry from changes in Triangular Irregular Networks (TINs). While this manuscript is within the scope of GMD and presents a novel approach to the classification of faults, I feel that there are substantial changes required to clarify assumptions, methods and results. For example, the key assumption that faults at the surface of the Earth are reflected by particular landforms (e.g. scarps or breaks in slope) and their geometry is not entirely valid nor is it clearly stated. | **Clarification**

This is a misunderstanding. We didn't analyze faults at the surface of the Earth. We used borehole (subsurface) data with preferred orientation of the surface. We had this information in the section *Real data*:

"we used borehole data (Michalak, 2024a) corresponding to a horizon separating Middle-Jurassic rock units: Kościeliska sandstones from ore-bearing clay deposits"

"From a geometric perspective, KSH dips at low angles to NE"

However, we agree that the explicit formulation was missing and it is worth adding more information about the assumptions in other sections. | We added a figure to illustrate the applicability of the method.

A paragraph with explicit statements about the assumptions and data was added in the Introduction. |

|   |   |   |
|---|---|---|
|   |   |   |

A new figure illustrating the assumption of having subsurface data with preferred orientation:

[Figure]

2.

| Suggestion, Question, or Comment from the Reviewer #1 | Author's Response | Change in the Manuscript |
|---|---|---|
| The methods section should be structured to reflect the steps summarised in Fig 2 (which should be moved from the introduction to the methods section). | **Disagree**/Clarification

We are a bit confused about this request because Fig. 2 presents a triangle and its neighbors. So there are no specific steps illustrated in this figure. Probably you meant Fig. 3 because a comment of similar nature is written below (Comment #25).

If this is the case, we think that the specific sections in the Methods section properly reflect what is illustrated in Fig. 3. | None about this comment.

However, we followed the suggestions from comment #25 which points to the correct figure (Figure 3). Please note that the order of figures has changed. |

3.

| Suggestion, Question, or Comment from the Reviewer #1 | Author's Response | Change in the Manuscript |
|---|---|---|
| The results are ambiguous, given that there is confusion around what is being compared in the test data and how it is being compared. | **Clarification**

Actually, we don't compare anything in the test data. Test data are only used to evaluate the performance of the algorithm.

Maybe you meant comparing performance metrics before and after hyperparameter tuning? This information was provided in the captions to Table 1 and Table 2. But we decided to add one clarifying sentence in section 4.1.

Please note that our data sets are divided into:

- synthetic data: training data, test data

- real data (borehole data)

What we actually compared are the results from the unsupervised classification (presented in the paper Michalak et al. 2022) and the supervised classification (presented in the manuscript). | We added one clarifying sentence to the section 4.1 Synthetic data:

"However, the arbitrarily selected hyperparameters are not guaranteed to give the best performance of the algorithm. " |

References:

Michalak, M., Teper, L., Wellmann, F., Żaba, J., Gaidzik, K., Kostur, M., Maystrenko, Y., and Leonowicz, P., (2022). Clustering has a meaning: optimization of angular similarity to detect 3D geometric anomalies in geological terrains Solid Earth, 13(11), 1697-1720, https://doi.org/10.5194/se-13-1697-2022

4.

| Suggestion, Question, or Comment from the Reviewer #1 | Author's Response | Change in the Manuscript |
|---|---|---|
| Finally, the manuscript is hard to follow in many sections (see specific comments below) and the figures need a substantial amount of editing to clearly communicate to the reader | **Clarification**

We addressed specific comments below.

See our next responses. | Figures have been corrected. |

| | | |
|---|---|---|
| what they should glean from them (e.g. symbology, colours, quality). Unfortunately, for these reasons I am recommending that this manuscript be rejected. | | |

5.

| Suggestion, Question, or Comment from the Reviewer #1 | Author's Response | Change in the Manuscript |
|---|---|---|
| **Specific comments**

**1 Introduction**

Why is there a section called short summary? This information should be removed as everything is covered in the abstract | **Disagree**

We have a short summary because short summaries are mandatory for Geoscientific Model Development. Please see the below screenshots. | None. |

**Get ready**

Before submitting your manuscript for peer review, you are kindly requested to do the following:

- to decide which manuscript type is correct for your GMD manuscript, ensuring that you can fulfil the requirements outlined on the manuscript type page. For most paper types, it is required to make model code available. We recommend that your manuscript includes the model code or that the code is submitted to a reliable repository and linked from your manuscript through a DOI. Please see our guidelines on code and data policy.
- to decide whether your preprint, once accepted after access review, should be posted on EGU's preprint repository EGUsphere or in GMDD;
- to download the Copernicus manuscript templates for LaTeX or MS WORD, or to follow the instructions for R Markdown submissions;
- to prepare your manuscript *.pdf file according to the manuscript preparation guidelines including line numbers and page numbers. Should you have used AI tools to generate (parts of) your manuscript, please describe the usage either in the Methods section or the Acknowledgements;
- to prepare your abstract text and your short summary, both to be pasted into the file upload form, as well as your supplement file as *.pdf or *.zip archive, if applicable;

**Highlight articles**

30 Apr 2024
NEWTS1.0: Numerical model of coastal Erosion by Waves and Transgressive Scarps
Rose V. Palermo, J. Taylor Perron, Jason M. Soderblom, Samuel P. D. Birch, Alexander G. Hayes, and Andrew D. Ashton
Geosci. Model Dev., 17, 3433–3445, https://doi.org/10.5194/gmd-17-3433-2024, 2024
▸ Executive editor

[Figure]

Short summary
Models of rocky coastal erosion help us understand the controls on coastal morphology and evolution. In this paper, we present a simplified model of coastline erosion driven by either uniform erosion where coastline erosion is constant or wave-driven erosion where coastline erosion is a function of the wave power. This model can be used to evaluate how coastline changes reflect climate, sea-level history, material properties, and the relative influence of different erosional processes.
▸ Hide

6.

| Suggestion, Question, | Author's Response | Change in the Manuscript |
|---|---|---|

| or Comment from the Reviewer #1 | | |
|---|---|---|
| The use of the term geological terrains is a little misleading. The model uses changes in slope in triangular irregular network (TIN) models, presumably of the Earth's surface (this is not clear), as the basis for detecting faults. | **Agree/Clarification**

We agree that the term "geological terrains" can be better replaced with "triangular irregular network (TIN) models".

However, it is not true that we investigate changes in the slope of the Earth's surface. We used subsurface data sets. The surfaces are assumed to have preferred orientation (a sequence of conformal units). | We have changed the title to better reflect that we work with triangulated models of the terrain.

A clarification about assumptions has been added in the Introduction. |

7.

| Suggestion, Question, or Comment from the Reviewer #1 | Author's Response | Change in the Manuscript |
|---|---|---|
| What if for instance, the user has a digital elevation model? Would they still be able to use this approach given that the surface model is not irregular? | **Clarification**

These are potentially interesting questions, however they again result from misunderstanding that we use data points from the Earth's surface. We note that we use subsurface borehole data (section 4.2 *Real data*). For the subsurface you could potentially obtain high resolution data (e.g. via seismic methods), however I believe that most researchers would disagree that one can use the term "digital elevation model" for the subsurface because traditionally the term "digital elevation models" is reserved for representing the Earth's surface (or other planets).

In our study, we don't use regular data, so we cannot effectively communicate the answer to the question about applicability of data points in regular networks. However, | None. |

| | we cannot see any formal objection preventing the use of regular data sets. We believe that the assumptions about the terrain geometry are more important than the spatial arrangement of input points. As such, we would expect that the proposed fault detection method should work for regular data point sets provided that the data points represent a terrain with preferred orientation (e.g. a homocline being part of a syncline). | |

8.

| Suggestion, Question, or Comment from the Reviewer #1 | Author's Response | Change in the Manuscript |
|---|---|---|
| One key assumption that I don't think has been stated clearly enough is that this approach assumes that all scarps, and/or breaks in slope are caused by faulting, i.e. they are fault-controlled landforms. This is probably not the case as some landforms will be controlled by other geological processes and features such as erosion and variability in rock type resistance to weathering. Furthermore, not all faults will be reflected as changes in the landscape. | **Clarification**

We agree that for some datasets some of the identified anomalies can be attributed to erosion. In particular, this may relate to Earth's surface data.

However, in our study, we used borehole data that documented a subsurface geological interface with preferred orientation (from a sequence of conformal units). So it is less likely that the lineaments are caused by erosion.

We admit that there is a possibility that even in our case some of the identified lineaments can be attributed to erosion due to a hiatus related to the lack of an Ammonite zone but there is no research available about landforms developed in this time period.

In the ideal example, one should have a continuous sequence of uniformly | In the Discussion, we added a subsection ''5.3 Complexities of real data'' |

| | oriented layers. In such conditions, it is less likely that changes in the morphology of the interfaces are caused by erosion. However, we agree that the assumption should be more clearly presented. | We added a new figure in the Introduction and a paragraph with the statements about assumptions. |
|---|---|---|

9.

| Suggestion, Question, or Comment from the Reviewer #1 | Author's Response | Change in the Manuscript |
|---|---|---|
| **2 State of the art**

Supervised machine learning has been applied to a multitude of applications other than lithology classification, and I am not sure how relevant these applications are to your example, which is specifically the linking of TIN segments based on their location and normal and dip vectors. | **Clarification**

While we agree that "lithology classification" is not directly related to the study objective, we believe that it is within the more general scope of "geological mapping" which is also applicable for our study. Moreover, mentioning such studies can be useful for Editors to find referees with expertise in machine learning because the adoption of machine learning methods in structural geology is progressing but still it is not very much popular.

Please note that "lithology classification" was only one example in our State of the Art (now Background) section. We had more relevant examples of using machine learning in fault detection problems. The references are given below. | None. |

References

An, Y., Guo, J., Ye, Q., Childs, C., Walsh, J., and Dong, R.: Deep convolutional neural network for automatic fault recognition from 3D seismic datasets, Comput. Geosci., 153, 104776, https://doi.org/10.1016/j.cageo.2021.104776, 2021

Kaur, H., Zhang, Q., Witte, P., Liang, L., Wu, L., and Fomel, S.: Deep-learning-based 3D fault detection for carbon capture and storage, Geophysics, 88, IM101--IM112, https://doi.org/10.1190/geo2022-0755.1, 2023

Mattéo, L., Manighetti, I., Tarabalka, Y., Gaucel, J. M., van den Ende, M., Mercier, A., Tasar, O., Girard, N., Leclerc, F., Giampetro, T., Dominguez, S., and Malavieille, J.: Automatic Fault Mapping in Remote Optical Images and Topographic Data With Deep Learning, J. Geophys. Res. Solid Earth, 126, https://doi.org/10.1029/2020JB021269, 2021

10.

| Suggestion, Question, or Comment from the Reviewer #1 | Author's Response | Change in the Manuscript |
|---|---|---|
| **3 Methodology**

The training data are synthetically generated based on user inputs (there is a list of these at lines 121-124). Please mention that the training data are synthetic in line 120 and consider providing the user defined variables as a table, for example, with an indication of the parameter name (as in the application) the range of possible values (surely there will be some parameters with restrictions on numeric values, e.g. non-negative) and the function of the parameter. This table may be useful to document other input parameters for the application if there are any. | **Agree/ Clarification**

We agree that mentioning synthetic data and providing a table with the parameters with ranges can be useful. We are a bit confused, however, what you mean about "function of the parameter". Do you mean the name of the function in the code? There are no specific functions in the code for reading the parameters from the user. But since you say that the structure of the table is only an example, we decided that our last column will point to the names of the variables in the computer code. | • We added the word "synthetic".

• a table with parameters was added to the manuscript (Methods) |

11.

| Suggestion, Question, or Comment from the Reviewer #1 | Author's Response | Change in the Manuscript |
|---|---|---|
| The structure of the paragraphs for generating training data is hard to follow. It reads better | **Agree**

We agree that it can read better using numbered dot points. | We introduced the numbered dot points. |

| | | |
|---|---|---|
| with numbered dot points for each of the steps. For example,

The faulted triangulated terrains are created in the following sequence (summarized in Fig. 5).

1. a container with 2D points is generated within a square of a given size**.**

2. a new container of 3D points is created with the Z coordinate corresponding to the random value of dip and dip direction.

3. noise is introduced to the surface defined as a random fraction of the elevation difference within the terrain.

4. … | | |

12.

| Suggestion, Question, or Comment from the Reviewer #1 | Author's Response | Change in the Manuscript |
|---|---|---|
| You must make sure that the table with user defined parameters has the same names as the parameters included in these steps to avoid ambiguity. | **Agree** | A table with user defined parameters was prepared. The names of the parameters in the table match the names of the parameters in the text. |

13.

| Suggestion, Question, | Author's Response | Change in the Manuscript |
|---|---|---|

| or Comment from the Reviewer #1 | | |
|---|---|---|
| Lines 150-153: it is unclear if dip direction is included in the final set of variables for supervised learning. The statement "northern directions indicate great numerical difference (e.g. 358-2=356) but very small geometric difference (4 degrees)." Needs to be clarified as I suspect that you trying to explain that the orientation difference between 358 deg and 002 deg (as measured relative to magnetic/grid north) is 4 degrees but numerically it is 356. | **Clarification**

No, dip direction is not used as a feature for classification. This paragraph was actually a justification why dip direction should not be used for classification in our case. | We added a sentence saying that we don't use dip direction for classification. |

14.

| Suggestion, Question, or Comment from the Reviewer #1 | Author's Response | Change in the Manuscript |
|---|---|---|
| Line 169: The authors state that "sort the distances to neighbouring triangles in decreasing order." to avoid randomness issues. Which distance or distances are used to sort the neighbouring triangles? Is this the Euclidean distance or cosine distance of the normal and dip vectors or is this something else? Please clarify. | **Disagree/Clarification**

We are not not computing distances of vectors (because distance for an individual vector does not exist) but distances between vectors. We only compare dip vectors with dip vectors and normal vectors with normal vectors. There is no mixing of normal vectors and dip vectors in these comparisons (distance measurements).

Actually, distances between a triangle and its neighbors are sorted. This applies to all three types of distance: angular, Euclidean and cosine. The distances are calculated either between | None. |

| | normal vectors of a triangle and its neighbors or between dip vectors of a triangle and its neighbors.

For example, for a triangle $t$ and its three neighbors you can have three distances as follows:

$d(t, n_1)=0.4$,

$d(t, n_2)=0.1$,

$d(t, n_3)=0.7$.

And you just sort them: 0.7, 0.4, 0.1. And now we are able to rearrange the order of neighbors: so the first neighbor is the one with the highest value of distance equal to 0.7, the second neighbor is the one with the middle value of distance (0.4) and the third one is the neighbor with the smallest value of distance.

This sorting is necessary because initially the order of neighbors corresponds only to a counterclockwise order (see References). So without sorting, for a sequence for triangles, their counterclockwise-ordered neighbors can have no consistent geographical position relative to the selected triangle. For example, for the first triangle its first neighbor can be to the North, and for the second triangle, its first neighbor can be to the South. So without sorting, we have the potential to confuse the algorithm. | |
| --- | --- | --- |

References (about counterclockwise order of neighbors):

Getting started with CGAL, https://graphics.stanford.edu/courses/cs368-04-spring/manuals/CGAL_Tutorial.pdf , p. 31

15.

| Suggestion, Question, or Comment from the Reviewer #1 | Author's Response | Change in the Manuscript |
|---|---|---|
| Line 176: Authors state that visualsaion uses spatial clustering. Not sure what you mean by spatial clustering as I cannot see any indication of a specific spatial clustering approach. XXXX Appears to be more like the spatial distribution of classes (fault or not fault) as plotted on a map | **Clarification**

We believe that the method that we applied matches the definition of "spatial clustering" as defined by Fisher:

"studying the directional aspect of the data separately from the spatial aspect, and then to put any resulting clusters of directions back into their spatial context" (p. 193)

Indeed, the SVM performs the classification without taking into account spatial information (it uses only geometric information). And then, the resulting labels (-1 and 1) of classification are displayed on the map.

We don't know what XXXX means. | We added more information about spatial clustering to the manuscript. |

References: Fisher, N.I., 1993. Statistical Analysis of Circular Data. Cambridge University Press.

https://doi.org/10.1017/cbo9780511564345

[Figure]

Fig. 7.15 (*cont.*) (c) Smoothing using 25% of data. See Example 7.9.

lineaments can be swung quite dramatically in a smoothed version of the data. Weighting is generally available in smoothing algorithms.

**Example 7.9** For the lineament data displayed in Figure 7.3, it is appropriate to examine subsets in different arcs. Figure 7.13 shows the data in the 45° arc centred on the NE–SW axis. Some patches are evident in which this lineament set is under-represented. In Figure 7.14, two smoothed version are displayed, based on *loess* using spans of 10% and 25%. The more heavily smoothed version in Figure 7.14(b) seems to be revealing some tendency for the orientations to exhibit some systematic rotation from site to site across the region. This sort of exploratory analysis can be carried out for a set of arcs spanning the complete range, and the various bits of information combined. It is also helpful to explore the data set as a whole. Figure 7.15 shows three smoothed versions of the complete lineament set using length-weighting. The most lightly smoothed version (Figure 7.15(a)) clarifies the patterns locally, with the shortest lineaments altered slightly so that they conform more closely to their neighbours; the most heavily smoothed (Figure 7.15(c)) brings out gross predominant trends, although at the cost of rotating at least one major lineament substantially, so that the result may be misleading. Figure 7.15(b) preserves some local structure as well as revealing larger-scale patterns. The next step would be to relate the orientations to the known geological history of the region.

**References and footnotes.** The spline method of Watson (1985) (see also Mendoza 1986) is an alternative form of smoothing. A general discussion of the spline method and methods based on local regression can be found in Hansen & Mount (1990), in the context of smoothing crustal

[Figure]

Fig. 7.16 Summary of fracture data from a colliery, after the data have been grouped into grid cells, then summarised in terms of the directions and strengths of the modal groups in each cell.

stress orientation measurements. Young (1987a, 1987b) has used a kriging approach to spatial smoothing of unit vectors and axes.

**(ii) Spatial clustering.** As a simple example of the sort of problem we have in mind, consider a region which has been affected by three significant fracturing events. The first and second of these may have created two ubiquitous sets of fractures trending N–S and E–W, say. Subsequently, a relatively narrow strip which passes through the centre of the region is affected by the third fracturing event trending NE–SW. The region is thus divided into three *structural domains*, within each of which the fracture pattern is homogeneous. Two major objectives of domain analysis are to identify the number of homogeneous structural domains and to determine which are similar.

One way of identifying spatially non-contiguous areas of similar fracture pattern is to study the directional aspect of the data separately from the spatial aspect, and then to put any resulting clusters of directions back into their spatial context. As an example of how this might be done, the
* * *
Fig. 7.17 A display resulting from performing a Principal Coordinates' Analysis (PCA) on the summary cell diagrams, or 'birds' feet', in Figure 7.17, using the similarity measure $S(P, Q)$. Each bird's foot has been plotted out in the $x$–$y$ space defined by the first two principal coordinates.

analysis of Fisher *et al.* (1985) will be described briefly: for fuller details, the reference should be consulted. (The data set considered in that paper consisted of fractures in the tunnel roof throughout a coal mine, with fractures tending to be of comparable length. By way of comparison, the fracture data in Example 7.3 have a wide range of fracture lengths; however, for the grid-based method below, this difference is less pronounced than it might be for other methods.)

There are three basic steps to the procedure:

**(1) Summarisation.** A grid is placed on the data map. For each grid cell, the trends of all fractures intersecting that grid cell are recorded. Denote the number of non-empty grid cells by $N$. The sample of measurements in each cell is then analysed to identify the number of modal groups, and the relative size and frequency of each. For a typical cell in which, say, $m$ modal groups have been identified, denote the summary information by $\{(\bar{\theta}_i, p_i), i = 1, \ldots, m\}$, where $p_i$ is the proportion of the cell data in the modal group with mean trend $\bar{\theta}_i$. A simple summary plot for this cell is then

Fig. 7.18 A preliminary grouping of the birds' feet in Figure 7.17, by a geologist.

furnished by a plot of the $m$ mean trends $\bar{\theta}_i$ as vectors of length $p_i$ radiating from some common origin. The complete data set can now be displayed in summary form by plotting these $N$ individual 'birds' feet' at the centres of their respective grid cells. Such a graph for the colliery data in Fisher *et al.* (1985) is shown in Figure 7.16.

**(2) Clustering directional summaries based on angular similarity.** It is not a straightforward matter to decide whether two birds' feet are similar, given that they might well have different numbers of trend vectors of varying lengths. A lengthy discussion of this point, together with details of an experiment to evaluate a number of possible measures, is given in the reference. The following *ad hoc* method (including formulae (7.28)–(7.31)) was found to give the best results in a test experiment.

Let $P = \{(\bar{\theta}_i, p_i), i = 1, \ldots, m_1\}$ and $Q = \{(\bar{\phi}_i, q_i), i = 1, \ldots, m_2\}$ be two summary sets, where $m_1 \le m_2$, say, and consider all possible match-ups of the $m_1$ mean trends $\bar{\theta}_1, \ldots, \bar{\theta}_{m_1}$ with $m_1$-subsets of $\bar{\phi}_1, \ldots, \bar{\phi}_{m_2}$. We shall define as *optimal* the particular match-up

$$(\bar{\theta}_1, \bar{\phi}_{j_1}), \ldots, (\bar{\theta}_{m_1}, \bar{\phi}_{j_{m_1}})$$

16.

| Suggestion, Question, or Comment from the Reviewer #1 | Author's Response | Change in the Manuscript |
|---|---|---|
| Lines 210-215: please format the equations for precision and recall and F1 such that they are on separate lines from the text. | **Agree** | Done. |

17.

| Suggestion, Question, or Comment from the Reviewer #1 | Author's Response | Change in the Manuscript |
|---|---|---|

| 4 Results | Clarification | |
|---|---|---|
| How many samples in the synthetic training data and how many samples are in the synthetic test data? What were the parameters used to generate the synthetic data for this experiment? | There are 1000 files and every file contains 100 input points (3D points). As such every file will have a bit less than 200 triangles (according to the well-known theorem linking vertices with faces of the triangulation - see below). So there are a bit less than 200 000 triangles. We also performed cleaning: removing collinear configurations, and removing triangles with the infinite face as one of their neighbors. Therefore 145297 triangles are left. | Information about the number of samples was added to the Result section (4.1). |
| | And only a small fraction of triangles are fault-related triangles (12411). Therefore, to reduce class-imbalance, we randomly select 12411 observations from the class with non-fault observations. | |
| | Taking all considerations into account, we have 12411 observations for each class (-1 and 1). | |
| | The parameters can be found in the uploaded data set (Zenodo). However, we provide the information here, as well: | A Table with the parameters used in this study was added to the manuscript (section 3.1). |
| | Number of files: 1000 | |
| | Terrain size: 1 | |
| | Dip angle: 0.5-2 | |
| | Dip direction: 20-70 | |
| | Number of points in the triangulation: 100 | |
| | Noise of the surface: 0.02-0.04 | |
| | Fault throw: 0.05-0.25 | |
| | The meaning of each parameter is described in the section 3.1 Generating terrains. | |

References: De Berg, M., Cheong, O., Van Kreveld, M., and Overmars, M.: Computational Geometry: Algorithms and Applications, 3rd Ed., Springer, 364 pp., https://doi.org/10.2307/3620533, 2008.

This is made precise in the following theorem.

**Theorem 9.1** *Let P be a set of n points in the plane, not all collinear, and let k denote the number of points in P that lie on the boundary of the convex hull of P. Then any triangulation of P has $2n - 2 - k$ triangles and $3n - 3 - k$ edges.*

*Proof.* Let $\mathcal{T}$ be a triangulation of $P$, and let $m$ denote the number of triangles of $\mathcal{T}$. Note that the number of faces of the triangulation, which we denote by $n_f$, is $m + 1$. Every triangle has three edges, and the unbounded face has $k$ edges. Furthermore, every edge is incident to exactly two faces. Hence, the total number of edges of $\mathcal{T}$ is $n_e := (3m + k)/2$. Euler's formula tells us that

$$n - n_e + n_f = 2.$$

18.

| Suggestion, Question, or Comment from the Reviewer #1 | Author's Response | Change in the Manuscript |
|---|---|---|
| It would be great to present the evaluation and validation data as a confusion matrix. Precision and recall can be appended to these tables. | Agree. | We added confusion matrices before and after hyperparameter tuning (Tables 3 and 5). We modified the computer code to generate confusion matrices before and after hyperparameter tuning. |

19.

| Suggestion, Question, or Comment from the Reviewer #1 | Author's Response | Change in the Manuscript |
|---|---|---|
| Please change Tab to Table where it occurs. | Agree. | Done. |

20.

| Suggestion, Question, or Comment from the Reviewer #1 | Author's Response | Change in the Manuscript |
|---|---|---|
| I am a little confused about how borehole data and the fault models based on the analysis of topographic features can be compared? Please explain this more clearly. Also you need to include at least a confusion matrix of the comparison. If it is not a quantitative comparison then I suggest that you exclude this. | **Clarification**

We are not sure but this comment can again be a result from misunderstanding that we use Earth's surface data.

To better understand the workflow of the study we note that it is divided into following stages:

● generation of synthetic faulted terrains based on random point generators (they can be considered synthetic borehole data) ,
● training the classification algorithm SVM on synthetic data,
● evaluating the SVM on synthetic data
● investigating generalizability of the classification by evaluating the SVM on real data, for example borehole data.

In the Result section we also studied the added value of using the supervised version compared to the unsupervised version used in our previous papers (Michalak et al. 2022). The conclusion from this comparison is that we were able now to find more types of faults, specifically we were able to detect faults trending perpendicular to | We added confusion matrices before and after hyperparameter tuning. |

| | the preferred dip direction of strata. | |
| --- | --- | --- |

References:

Michalak, M., Teper, L., Wellmann, F., Żaba, J., Gaidzik, K., Kostur, M., Maystrenko, Y., and Leonowicz, P., (2022). Clustering has a meaning: optimization of angular similarity to detect 3D geometric anomalies in geological terrains Solid Earth, 13(11), 1697-1720, https://doi.org/10.5194/se-13-1697-2022

21.

| Suggestion, Question, or Comment from the Reviewer #1 | Author's Response | Change in the Manuscript |
| --- | --- | --- |
| It seems that the only measure of success is fault or not fault and there is no measure of the successful classification of the orientation of the faults. Have you considered this as measure of fit for your classification model? | **Clarification**

In the literature the fault detection problem was always posed as a binary classification problem. And for every binary classification problem you need variables used for training and target labels. The confusion matrix and the metrics resulting from the confusion matrix such as precision and recall are standard for evaluating performance of the algorithm. As such, we don't use the orientation of the faults as a measure of the successful classification. Maybe it could be used for regression problems where the goal is to predict the fault orientation but regression problems are different from classification problems. | None. |

22.

| Suggestion, Question, or Comment from the Reviewer #1 | Author's Response | Change in the Manuscript |
| --- | --- | --- |
| **5 Discussion**

I have read this section several times and I am still a little confused. I suspect that there | **Agree/ Clarification**

We agree that separating Discussion into separate sections can be useful. | |

| Suggestion, Question, or Comment from the Reviewer #1 | Author's Response | Change in the Manuscript |
|---|---|---|
| are several aspects that you are trying to discuss:

● the use of TINs means that there are only a limited number (3) neighbours to every face and that this simplifies the modelling

● The assumptions when generating synthetic training data, e.g. planes representing faults

● Issues with multiple faults being predicted from a single synthetic fault training example that have different dips (although I am not entirely sure if I understand this correctly)

I suggest a careful review of the discussion with the view to clearly distinguish the main points (as sub sections of the discussion with headings) and clarify to the reader the key message for each of the points. | We will now discuss the questions raised in the points:

● The use of TINs means that there is a constant number of features for observations (except those that are at the edge of the convex hull but these are removed). This is not very much about simplifying the modelling (we note that modelling is simplification by definition) but making it actually valid.
● One of the major assumptions that we discuss is that we don't allow points to lie on the fault surface.
● Regarding ''multiple faults'': The problem is that for wide fault zones with points lying on the fault surface, we could get two sequences of fault-related points instead of one sequence. | We divided the Discussion into separate sections. |

23.

| Suggestion, Question, or Comment from the Reviewer #1 | Author's Response | Change in the Manuscript |
|---|---|---|
| **6 Conclusions**

The conclusion should clearly state, what you did (developed a supervised fault classifier) and how it is novel and | **Agree/ Clarification**

We added more fundamental information of this kind to Conclusion. In particular, we added that the | Conclusion section has been modified, accordingly. |

| | | |
|---|---|---|
| different from other approaches (generate synthetic terrain data representing faults that control landscape geometry, use of TINs to simplify modelling) and the key assumptions of the method. You should also communicate the impact of your work (who should be using your fault classifier and why). | terrains should have preferred orientation.

We believe that the classification approach can be used by geologists interested in geological complexity of subsurface environments with preferred orientation and limited availability of data. | |

24.

| Suggestion, Question, or Comment from the Reviewer #1 | Author's Response | Change in the Manuscript |
|---|---|---|
| **Figures**

Figure 1 - What do the colours in A represent? The legend states scalars but it is not clear what the scalars are, I suspect it is distance above some reference? | **Agree/ Clarification**

Colours in A (scalars) represent elevations. | We improved the following figures:

● Figure 1 (presenting a single faulted terrain and three example terrains)
● Figure 5 (workflow of generating synthetic terrains) |

The improved Figure 1 looks as follows: (elevation instead of scalars)

[Figure]

We also improved the old Figure 5 (now Fig. 6):

[Figure]

25.

| Suggestion, Question, or Comment from the Reviewer #1 | Author's Response | Change in the Manuscript |
|---|---|---|
| Figure 3 - This figure needs to be moved to the Methods section where the steps are summarised in detail. Likely introduced in an initial paragraph before section 3.1. At the moment this workflow lacks this clarifying information. | **Agree** | We moved Figure 3 to the Methods section. Please note that its number has changed now.

We added an initial paragraph before section 3.1.

We decided to move the sub-section about Visualization (now Spatial clustering) to the end of the Methods section. |

26.

| Suggestion, Question, or Comment from the Reviewer #1 | Author's Response | Change in the Manuscript |
|---|---|---|
| Figure 4 - It is unclear what is being presented here. Is this a 2D view of an underground mine or mining region? What | **Agree/ Clarification**

Figure 4 presents a subsurface geologic horizon restricted to the area studied | We added more information in the caption such as: |

| data are used to generate the points in 4B and 4C? What clustering algorithm used and what are the variables used in clustering? It appears that this information is provided in Michalak et al. (2022). I realise that a certain level of knowledge is assumed but for someone who has not read previous iterations of this research need to be provided with more background knowledge. It is probably worth indicating that this figure is modified from | in our manuscript. Figure 4 is composed of three panels:

a) faults identified by miners within a displaced ore-bearing clays horizon,

b) results from clustering dip vectors to Delaunay triangles using k-means with two clusters,

c) results from clustering dip vectors to Delaunay triangles using k-means with three clusters. We will now answer the questions.

● What data are used to generate the points in 4B and 4C?

We used borehole data (3 D points) from a displaced geologic interface between two geologic units. Then, we used Delaunay triangulation and the points in the figure represent geometric centres of Delaunay triangles.

● What clustering algorithm used and what are the variables used in clustering?

The k-means algorithm was used and the variables were either normal vectors to triangles or dip vectors defined as the projections of the normal vectors onto the triangle's plane. | ● algorithm used for unsupervised learning
● we used the same borehole data in the unsupervised version
● the figure is a modified version of the figure from a paper already published |

27.

| Suggestion, Question, or Comment from the Reviewer #1 | Author's Response | Change in the Manuscript |
|---|---|---|
| Figure 6 -I am having trouble seeing the differences between | **Agree/Clarification** | |

all of the plots in Fig 6. Is there some way that you can change the shape or the colour of the points in each of the models and indicate how these points compare with the borehole data or the unsupervised model, whichever is being compared in this figure?

It is true that it is difficult to see the differences. There may be two reasons:

- results are not very much sensitive to changing hyperparameters of the algorithm.

-there was a bug in the code and panel a) was repeated on panel c). It has been corrected.

I marked with a green circle an example area where the classification results are different for every plot.

We agree that it can be a good idea to enhance differences between the results for supervised classification and results from the unsupervised model.

We corrected the Python notebook because there was a bug in the creation of panels for real data.

Old (bad) version:

```
training_mono.insert(2
,
"svm_configuration2",
predictions_mono)
```

Correct version:
```
training_mono.insert(2
,
"svm_configuration2_co
rrected_2",
predictions_mono_confi
guration2)
```

On the panel (a), we marked lineaments known from the unsupervised learning revealed in previous studies (Michalak et al., 2022). The new lineaments revealed in this study using the supervised classification and the area where both groups intersect are marked, as well.

Improved figure illustrating the added value of supervised classification compared to the unsupervised classification.

[Figure]

28.

| Suggestion, Question, or Comment from the Reviewer #1 | Author's Response | Change in the Manuscript |
|---|---|---|
| Figures 7 and 8 - The symbols in these figures are hard to see as they are very small. Also these figures can probably be combined into a single figure as A and B. | **Agree** | Done. |

**Responses to Reviewer #2**

1.

| Suggestion, Question, or Comment from the Reviewer #2 | Author's Response | Change in the Manuscript |
|---|---|---|
| General comments

The manuscript "Broken Terrains v. 1.0: A Supervised Detection of Fault-Related Lineaments on Geological Terrains" introduces a novel machine-learning approach but challenges readability, making it difficult to follow the progression of ideas. For example, the section on geological settings is well-contained within a single paragraph. Still, whether the subsequent text belongs to this section or would be more appropriately placed in the Results or Discussion sections. The manuscript would benefit significantly from a comprehensive restructuring to enhance coherence and flow. Additionally, the figures require careful editing to improve their visual impact; for instance, Figure 5 uses blue points on a deep grey background, a combination that lacks sufficient contrast and hinders clarity. Still, some technicalities need to be clarified, mainly how a detection model designed with synthetic 3D faults could be applied to borehole data. | **Agree/Clarification**

We would like to have the Geological setting composed of two sub-sections: one about regional information and the second about previous results.

This is because we have a large figure there showing results from using unsupervised learning. So if it was in the Results section, one could be confused whether this is a new result or an old one. But this is an old result which is shown in the manuscript to better see the added value of using supervised classification.

We agree that some of the figures required editing.

Regarding the question about synthetic data and borehole data: you can imagine that synthetic data are also (synthetic) borehole data. Actually, we believe that it is of less importance whether the data points are from boreholes or other sources. It is more important whether the points are sampled from a surface with preferred orientation. | We divided Geological setting into two sections:

● 2.2.1 Regional and geometric background
● 2.2.2 Discussion of previous results

Figure 5 (now Figure 6) has been corrected. Now, we have white points for better contrast.

We added a paragraph to Introduction that states the assumptions in a more explicit way. A new figure illustrating the assumption was added as well. |

Based on the comments from two Referees, we have improved the figure as follows:

- dark blue on panel A was replaced with white
- "elevation" instead of "scalars"
- additional legend to points on panel "f"

[Figure]

2.

| Suggestion, Question, or Comment from the Reviewer #2 | Author's Response | Change in the Manuscript |
|---|---|---|
| Specific comments

0.- Short Summary

This section does not look necessary to get the correct general idea of the manuscript, just like the words "to classify terrain shape or nearby features" when the main goal is fault detection. | **Clarification**

We agree that the words "to classify terrain shape or nearby features" may be confusing. However, we must have the Short summary because it is mandatory in GMD. | We have rewritten the Short summary to better reflect the intention that we detect faults with triangulated models. |

3.

| Suggestion, Question, or Comment from the Reviewer #2 | Author's Response | Change in the Manuscript |
|---|---|---|

| 1.- Introduction

line 38 "lineament/fault" is not recommended to use the "/" in formal manuscripts. The training set of Figure 1 looks very similar and has short faults but is rotated in different 3D positions. It looks like quite a simple idealized model. Could you add more complexity, such as fault displacement variation?? In Figure 3 and general, it is better to be specific with quantities instead of using the term "many." | **Clarification**

We agree that "/" can be avoided.

Regarding the old Figure 1 (now Fig. 2): this is only an illustration that we use triangulated terrains in the training data. Of course, we don't have only three terrains in the training data but one thousand terrains. Adding complexity in terms of fault displacement variation can be a good idea in future developments but in this study, we investigate if the simplest scenario works.

We agree about using quantities. | In most cases, we deleted "/" and replaced it with "or". In this particular case (l. 38), we decided to make it shorter: "to predict possible fault presence"

We replaced "many" with "one thousand" regarding the number of terrains. |

4.

| Suggestion, Question, or Comment from the Reviewer #2 | Author's Response | Change in the Manuscript |
|---|---|---|
| 2.- State of the art

I prefer to call this section "Background" instead of "State of the Art." | **Agree** | Done |

5.

| Suggestion, Question, or Comment from the Reviewer #2 | Author's Response | Change in the Manuscript |
|---|---|---|

| 2.2 Geological Setting

As I mentioned earlier, this section needs to be completed, and 105 paragraphs sound like a discussion instead of describing a geological terrain or setting. | **Clarification**

We would like to have the Geological setting composed of two sub-sections: one about regional information and the second about previous results.

This is because we have a large figure there showing results from using unsupervised learning. So if it was in the Results section, one could be confused whether this is a new result or an old one. But this is an old result which is shown in the manuscript to better see the added value of using supervised classification.

We agree that more geological information can be good for discussing Results. | We divided Geological setting into two sections:
● 2.2.1 Regional and geometric background
● 2.2.2 Discussion of previous results

We added more geological information in Discussion (5.3 Complexities of real data) |

6.

| **Suggestion, Question, or Comment from the Reviewer #2** | **Author's Response** | **Change in the Manuscript** |
|---|---|---|
| 3.- Methodology

3.1 .- Genereting terrains

It needs to be rewritten for clarity. | **Agree** | Section 3.1 has been rewritten. |

7.

| **Suggestion, Question, or Comment from the Reviewer #2** | **Author's Response** | **Change in the Manuscript** |
|---|---|---|

| 3.2 Selecting meaningful and consistent variables

I think that some terminology upgrades can be made here, like "variable features" or "feature" instead of just "variables," which could be general and prone to confusing terms. | **Agree/Clarification**

We would prefer to use the term "features". | We replaced "variables" with "features".

We first introduce the term "input variables" in the abstract and later on the term "features" is used, for brevity.

To avoid ambiguity with other applications and meaning of "features", we replaced the term "features" with "patterns" in the Discussion (5.4 Modelling assumptions"). |

8.

| Suggestion, Question, or Comment from the Reviewer #2 | Author's Response | Change in the Manuscript |
|---|---|---|
| 3.3 Visualization

This section is too short; eliminate or combine it with other sections. | **Disagree**

We decided to expand this section and rename it because the concept of "spatial clustering" (Fisher, 1993) is not only about visualization. It is also about investigating directional information separately from spatial information. The spatial information is added at the end when the clusters of directional information are identified. Our study can be considered a very specific version of the concept where clusters are identified using geometric information and supervised classification. | We added more information regarding the concept of "spatial clustering".

We replaced the section name "Visualization" with "Spatial clustering". |

References

Fisher, N. I.: Statistical analysis of circular data, Cambridge University Press, 277 pp., https://doi.org/10.1017/cbo9780511564345, 1993.

9.

| Suggestion, Question, or Comment from the Reviewer #2 | Author's Response | Change in the Manuscript |
|---|---|---|
| 4 Results.

In line 217, the parentheses are incorrectly placed. Additionally, there needs to be more essential details typically included in a machine learning approach, such as the number of samples used for training and testing and the number of correctly classified samples. The sections overall appear too brief; therefore, it would be beneficial to provide a more thorough description of the experiment to ensure that others can fully understand the methodology and results. | **Agree**

In our study, we used 1000 triangulated terrains with 100 points in every terrain. This configuration resulted in the initial number of 185980 triangles. We removed collinear configurations (collinearity>0.90) and triangles which did not have three finite neighbors. As a result, 145297 triangles remained. And only a small fraction of triangles are fault-related triangles (12411 vs 132886). Therefore, to reduce class-imbalance, we randomly select 12411 observations from the class with non-fault observations. Taking all considerations into account, we have 24822 samples with 12411 observations for each class (-1 and 1). Then, the set was divided into training (18616) and test (6206) set.

The below results were obtained before hyperparameter tuning (the sum relates to the number of samples in the test data):

2993 (true negatives)

243 (false positives)

123 (false negatives)

2847 (true positives) | Information have been added. |

10.

| Suggestion, Question, or Comment from the | Author's Response | Change in the Manuscript |
|---|---|---|

| Reviewer #2 | | |
|---|---|---|
| 5.- Discussion

This section is challenging to follow; I suggest it be rewritten for clarity. | **Agree**

We agree with Reviewer #1 and Reviewer #2 that this section can be divided into sub-sections. | Discussion section has been divided into sub-sections according to comments from Reviewer #1.

We also transferred the more "analytical" part of the Conclusion section to Discussion according to suggestions from Reviewer #2. |

11.

| Suggestion, Question, or Comment from the Reviewer #2 | Author's Response | Change in the Manuscript |
|---|---|---|
| 6.- Conclussions.

This section reads more like a discussion than a conclusion. Only lines 294 and 295 align with the intent of a conclusion. While the section is well-explained, I recommend relocating it to the discussion section. | **Agree** | We transferred the more "analytical" part of the Conclusion section to Discussion . |

---

## Referee Report (RR1)

This paper presents a method using support vector machines to identify faults cutting stratigraphic horizons when those horizons are represented by a triangular mesh created from scattered borehole data. The method is demonstrated on a case study and is compared to a clustering-based method developed in a previous paper by the same first author. The paper has undergone a previous round of review, in which unclear language emerged as a major issue, which the present revision seeks to address.

The method is novel to the best of my knowledge and is likely to be of interest to the geoscientific modelling community. The previous revisions have improved the clarity of the manuscript, in particular in regard to the usage of the word "terrains." However, I think that the paper still requires major revisions before it is ready for publication. Most of my comments regard the clarity of the text and figures, but I also have some questions about the use of seemingly redundant features in the classification. My detailed comments are below.

**Title**

The choice of "slopes" to replace "terrains" here and in the text is an improvement, but it still sounds odd to me. I am more likely to think of a slope as a topographic feature. I think a word such as "interfaces," "contacts," or "horizons" would sound better. I see that all three of these words are already used in some places in the text. (The example seems to be with a stratigraphic horizon, although I can see that the method could be applied to any planar interface.) If it is necessary to convey the "preferred orientation" part, perhaps this could be done by including a word such as "planar," "homoclinal," or "dipping." (For instance, one could say "homoclinal interfaces".)

**1 Introduction**

To help show the value of the study, I suggest including a sentence about the kinds of practical applications that this kind of fault identification would be useful for.

This section would benefit from a clearer statement of the problem. I think it could be something like this: Geological models are often created by interpolation of scattered borehole data. Because of the localized nature of boreholes, faults not intersected by them will be missed, and interpolation will create horizons that appear continuous across the faults. This paper proposes a way to identify the presence of faults in this situation.

Line 29: The comparison to seismic data is good, but I don't think it is clear what "subsurface topographic data such as subsurface slopes" means. I think the key difference is that this method works with scattered data, so I suggest phrasing this as a comparison of scattered data vs. seismic images.

Figure 2: The meaning of the line labeled "fault" and referred to as a "fault line" is unclear. This appears to be a 3D model, and a fault is a plane (or surface) in three dimensions. If this is meant to show the fault, it should be shown as a plane. If this is meant to show the line of intersection between the fault and the "slope," then it should be labeled and described as such.

**2 Background**

Since the paper involves significant comparisons to Michalak et al. (2022) and uses the same example as them, I think it would be helpful to divide this section into two paragraphs: One discussing other machine learning applications more generally, and the other specifically presenting Michalak et al. (2022) as a preceding work that this paper will build on.

**3 Methods**

Figure 4: I find the subfigures on the left of this figure confusing. I assume they correspond to the steps of the flow chart on the right, but it would help to state that explicitly in the caption and perhaps connect them within the figure. Also, it would help to give them letters and refer to those letters in the caption.

Also in Figure 4: Would training the model also be a step in the workflow? Or is that meant to be included in the "evaluation for synthetic data" step? If the latter, perhaps it could be "training and evaluation with synthetic data."

Lines 151-152: The use of normal and dip vectors seems redundant. Either one can be used alone to define the orientation of the plane, and either one can be derived from the other. Please either explain why both are needed or try doing the analysis using just one of them.

Lines 151-159: The different distance types also seem somewhat redundant. While they will give different numbers, they are all closely related. For instance, the square of the Euclidean distance is proportional to the cosine distance as discussed by Michalak et al. (2022). Please explain why three different distance measures are used, rather than just one as in Michalak et al. (2022).

Line 153: It would be helpful to give the variable names after the distance types: "angular distance ($d_a$), Euclidean distance ($d_e$), and cosine distance ($d_c$).

Line 161: Instead of "middle part of Fig. 4", I think it would be clearer to give this subfigure a letter (such as Fig. 4b) and refer to that.

Line 167-172: This is, indeed, a good reason to use vector representations of orientations, rather than dip and dip direction. However, this discussion doesn't seem to fit with the preceding part of the paragraph. I would suggest either making it a separate paragraph or moving it to where you first talk about using vectors.

*Tables 2 and 3*: Dip angle and dip direction in these tables appear to be in degrees, so they should have the degree symbol (°).

*Figure 5:* What is the meaning of the green box and letter ε in part c? Also, in the caption for part f, I think that "fault line" should be "fault plane" (see also my comment on Figure 2).

**4 Geological Setting**

I don't think it is necessary to divide this section into two subsections when they are only one paragraph each.

Lines 232-233: Saying the KSH is a "geological unit" sounds to me like it is a specific rock unit within the stratigraphy. But given that it is a homocline, I assume it is a structure. If that is correct, I suggest changing this to "a geological structure." Also, if the homocline forms one side of the synclinorium, I would suggest changing "a slope" to "a limb" in Line 232.

Line 234: It would be helpful to give a range of values to quantify "low angles."

**5 Results**

Lines 272-276: For the grid search optimization, it would be helpful to state the range of values tested and the grid spacing for each parameter.

Lines 291-293: I suggest moving the information about the specific horizon to Section 4 to go with the rest of the discussion about the case study.

**6 Discussion**

Line 329: Is the "unsupervised method" the one from Michalak et al. (2022)? It would be clearer if that were stated explicitly, such as by saying "the unsupervised method of Michalak et al. (2022)…"

Section 6.4: It appears to me that one assumption of the method is that the stratigraphy is homoclinal. That is a major modelling assumption that should be discussed.

Line 373: If it is "not clear" how the model would classify the structure in Figure 8, couldn't the model be tested on the example in Figure 8 to find out?

---

## Author Response (AR2)

In this file, there are responses to the Editor.

**Responses to the Editor**

We thank the Editor for his investigation about the root of the misunderstanding and the constructive feedback. We agree that our use of „terrains" and „lineaments" was misleading.

To better observe communicating our response, we divided our responses into three categories: Agree/Clarification/Disagree.

1.

| Suggestion, Question, or Comment from the Editor | Author's Response | Change in the Manuscript |
|---|---|---|
| I therefore conducted my own review of your revised manuscript in order to understand the cause for this discrepancy between your statements and those of the referees. I believe that the key to the misunderstandings may be language. In particular:
* A "lineament" is a geographical feature, i.e., one occurring at Earth's surface
* "Terrain" is a set of topography, also occurring at Earth's surface | **Agree**
We agree that the combination of words was misleading.
We think that the term „subsurface slopes" better aligns with the focus on subsurface topographic data with preferred orientation.
And it seems that the term „subsurface slope" is adopted well for the subsurface characterization studies (see the below reference). | We deleted the occurrences of „lineament" and „terrains" from the manuscript. |

**References:**

Du, J., Zhang, W. G., & Li, Y. (2021). Variability of deep water in Jordan Basin of the Gulf of Maine: Influence of Gulf Stream warm core rings and the Nova Scotia Current. *Journal of Geophysical Research: Oceans, 126*(5), e2020JC017136.

2.

| Suggestion, Question, or Comment from the Editor | Author's Response | Change in the Manuscript |
|---|---|---|
| For further consideration in GMD, please revise the paper to make it very clear what the input data are, how they are being classified, and where this is taking place. For example: Are we looking at data of elevations of buried stratigraphic contacts? Whatever it is, please be precise. | Clarification
Yes, we confirm that we are looking at data of elevations of buried stratigraphic contacts.
To avoid ambiguity: while the data are buried stratigraphic contacts, we don't use geographic data (such as elevation) in the classification. | We added information to the abstract and Introduction that borehole data are buried stratigraphic contacts. |

3.

| Suggestion, Question, or Comment from the Editor | Author's Response | Change in the Manuscript |
|---|---|---|
| I am afraid that your model name does not make sense to the broader community because it seems that it is not about terrain (surface topography) but rather about geological units observed in the subsurface. Perhaps you can find something clever! | **Agree**

As a new model name, we propose: „SubsurfaceBreaks" | The model name has been changed in the title.

The README file in the GitHub repository and Zenodo description have been revised.

Variable names in the C++ computer code have been changed: we replaced „terrains" with „subsurface_slopes''. |

4.

| Suggestion, Question, or Comment from the Editor | Author's Response | Change in the Manuscript |
|---|---|---|
| Section 3 requires a full rewrite. You should cleanly present your method to detect faults (your 3.3). | Clarification/ **Agree**

Our previous Method section was presented in a chronological order (from synthetic data, through data preparation to classification using Support Vector Machines and evaluation).

The requested rearrangement results in the engine of the task (Support Vector Machines) being more exposed. | The Methods section begins now with a short summary about the method used for classification and short presentation of variables.

Then, our previous section 3.3 (Support Vector Machines) is presented, as required. |

5.

| Suggestion, Question, or Comment from the Editor | Author's Response | Change in the Manuscript |
|---|---|---|
| Your section 3.4 might be relevant here as well for visualization. | Clarification/Disagree

We could do this. However, our initial section 3.3 is mathematical description of the Support Vector Machines. We believe that the section about visualization does not match this content very well. | None, as of now. The section about visualization is the last section. |

6.

| Suggestion, Question, or Comment from the Editor | Author's Response | Change in the Manuscript |
|---|---|---|
| Section 3.1 should appear after, and could even be given within the Synthetic Tests section. | **Agree** | Our previous section 3.1 appears after our previous section 3.3. |

7.

| Suggestion, Question, or Comment from the Editor | Author's Response | Change in the Manuscript |
|---|---|---|
| Section 3.2 appears to start at the middle of a stream of thought about how to add attributes to and select features. It is critical to place your new method at the forefront and to highlight what you have done and why: Which attributes do you use, and how do you do so? | **Agree**/Clarification

Our previous Method section was presented in a chronological order (from synthetic data, through data preparation to classification using Support Vector Machines and evaluation).

In the requested rearrangment, the Methods section starts with a very short summary about the classification tool, attributes and evaluation which, as we believe, satisfy the requirement to place our new method at the forefront.

Then, we present the Support Vector Machines algorithm as the engine of the classification which requires several paragraphs.

The description of the algorithm is followed by the description of the attributes and the preprocessing steps which is reflected by the rearrangement of the columns in a data frame. | We did the rearrangement of the Methods section.

We added a sentence clarifying that attributes are represented by columns in a data frame and sorting some of the features (distances) introduces their rearrangement in the data frame. |

8.

| Suggestion, Question, or Comment from the Editor | Author's Response | Change in the Manuscript |
|---|---|---|
| 1. In modern English, we take the Polish spelling of Kraków. No need to change the name of your home city when you change the language! | **Agree** | Done. |

9.

| Suggestion, Question, or Comment from the Editor | Author's Response | Change in the Manuscript |
|---|---|---|
| 2. Please note that the "short summary" stands outside of the main manuscript. | **Agree** | We deleted the „Short summary" from the main manuscript. |

10.

| Suggestion, Question, or Comment from the Editor | Author's Response | Change in the Manuscript |
|---|---|---|
| 3. Section 2.1 might not be necessary. A condensed version that need not go so into detail on the cited studies and focuses on relevance to the method presented here could be folded into other sections. | **Clarification**

If possible, we would like to have this section because there are many classification approaches for detecting faults. And it may be useful for a reader to decide if our method is relevant.

However, we understand that we should not go into detail in this section. Therefore, we propose to shorten this section. | We deleted from the section 2.1 some information about limitations of unsupervised approaches. |

11.

| Suggestion, Question, or Comment from the Editor | Author's Response | Change in the Manuscript |
|---|---|---|
| 4. You can move your case study to after the presentation of the method. | **Agree** | Done. |